# Decarbonization, population disruption and resource inventories in the global energy transition

Kamila Svobodova [1,2,3] ✉, John R. Owen [4], Deanna Kemp [1], Vítězslav Moudrý [3], Éléonore Lèbre [1], Martin Stringer [1] & Benjamin K. Sovacool [5,6,7]

We develop a novel approach to analysing decarbonisation strategies by linking global resource inventories with demographic systems. Our 'mine-town systems' approach establishes an empirical basis for examining the spatial extent of the transition and demographic effects of changing energy systems. The research highlights an urgent need for targeted macro-level planning as global markets see a decline in thermal coal and a ramp up of other mining commodities. Our findings suggest that ramping up energy transition metals (ETM) could be more disruptive to demographic systems than ramping down coal. The data shows asymmetry in the distribution of risks: mine-town systems within the United States are most sensitive to coal phase-out, while systems in Australia and Canada are most sensitive to ETM phase-in. A complete phase-out of coal could disrupt demographic systems with a minimum of 33.5 million people, and another 115.7 million people if all available ETM projects enter production.

Research into the impact of switching energy systems is vast. The rapidly increasing knowledge base includes understanding the current and future demand for energy products[1]; optimisation of supply patterns[2]; the impacts of innovation, technology deployment and diffusion on energy systems[3]; requirements for energy investments[4]; and energy policies related to efficiency, security, energy equity and justice[5–7]. Concerns about the equitable distribution of risk and benefit, as well as the future implications for workers and communities, are a central theme in the energy transition policy research[6,8].

Transition studies rightly focus on the market and policy process associated with shifting dependencies between a coal-based energy system and a system supported by renewable technologies[9–11]. Coal-focused research mainly relates to phase-out and the adjustments nations, markets and in some instances, specific population segments need to make in preparation[12–14]. These studies tend to be country or region-specific and centre on barriers to implementation, such as energy mix or economic dependency[15–17]. The mine closure literature is characterised by the absence of positive case examples, policies or planning instruments to manage social, economic and environmental problems at scale[18,19]. With the exception of individual case studies, there is no systematic assessment of phase-out effects at sub-national levels. Likewise, there are no publicly available global-scale assessments of coal inventories that link the location of mining projects to their immediate surroundings and the contingencies that form based on these locational attributes. A clear gap therefore exists between global resource inventories for coal and demographic systems, especially given the people-centred nature of current just transition debates[20,21].

Studies examining the role of renewables in the energy transition have a notably different research footprint. Scholars have utilised

[1]Sustainable Minerals Institute, The University of Queensland, Saint Lucia, QLD 4072, Australia. [2]Department of Agricultural Economics and Rural Development, University of Göttingen, 37073 Göttingen, Germany. [3]Faculty of Environmental Sciences, Czech University of Life Sciences Prague, Praha—Suchdol 165 00, Czech Republic. [4]Centre for Development Support, University of the Free State, Bloemfontein 9300, South Africa. [5]Science Policy Research Unit, University of Sussex Business School, Brighton BN1 9SL, UK. [6]Department of Earth and Environment, Boston University, Boston, MA, USA. [7]Center for Energy Technologies, Department of Business Development and Technology, Aarhus University, Aarhus, Denmark. ✉e-mail: k.svobodova@uq.edu.au

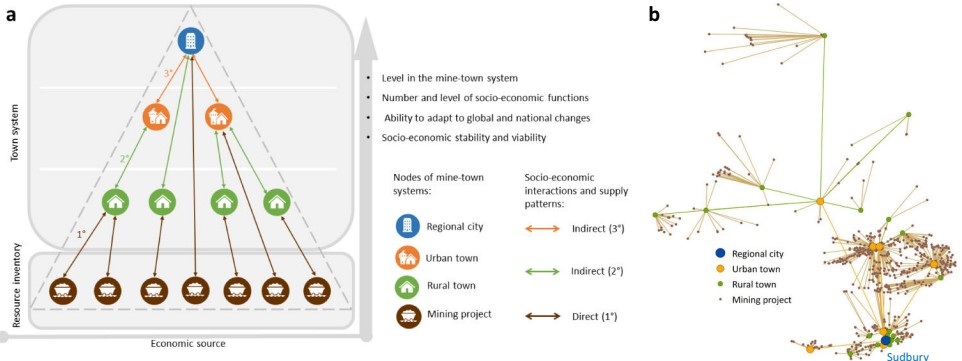

**Fig. 1 | The architecture of a mine-town system.** We define a mine-town system as a network of socio-economic connections between resource inventories and town systems. Mining projects and towns serve as 'nodes' in the mine-town systems. Mining projects are producers in the system and represent the resource inventory. The hierarchy of consumers (towns) reflects the different size and level of towns that are directly (1°) or indirectly (2° and 3°) dependent on mining projects as their economic source. As their position in the system elevates, the number and level of socio-economic functions increases along with the ability to adapt to global and national changes. **a** The definition of a mine-town system, **b** a real-world example of the Sudbury mine-town system in Canada.

resource inventories to examine constraints to mobilising geological materials for use in low-carbon technologies[22,23]. Criticality and availability studies, for example, assess the relative importance and relative ease associated with accessing these materials from a range of perspectives (environmental, fiscal, technological, geopolitical)[24–26]. Source risk assessments identify the scope of constraints around the development of resource projects based on local contextual factors and commonly known interactions that will occur between governance, environmental and social systems and the project[27–29]. Similar to coal-based studies, the research on energy transition metals, including those utilising resource inventories, does not establish or examine linkages between the location and type of resource project and the dependencies and contingencies attached to town systems.

Research on modelling of the energy transition largely overlooks its spatial context and dynamics[30–33]. However, the geography of the transition is becoming increasingly important[34–36]. A growing volume of research recognises spatially explicit sub-national processes critical for the switch of energy systems[37,38]. New theories are needed to inform leaders, scholars and citizens about the geographical complexity of the energy transition, specifically on how the transition unfolds across space, and where to plan and regulate the transformation.

We provide a novel contribution to energy transition policy debates by addressing the gaps identified above within the transitions studies, renewables and modelling literature. First, our work establishes a basis for mapping likely phase-out and phase-in dynamics of mineral resource extraction. Our work spans both sides of the energy transition by incorporating global resource inventories for coal on the one hand, and the energy transition metals required for renewable technologies on the other. Second, the spatial and demographic dimensions of the energy transition are assessed at a global scale, transcending the limitations of the existing literature. Novelty is achieved by linking the location and type of resource with the dependencies and contingencies attached to town systems and from there using these systems to assess dynamic interactions between resources and populations. In these systems, mining projects are 'drivers of change' and towns as 'agents of change'. Our mine-town systems connect mines to near population centres, and those centres to a hierarchy of larger demographic systems.

## Results

### Conceptualising mine-town systems

We develop a novel approach utilising the connectivity logic of the Food Web Model rather than the Central Place Theory which positions cities as centres of economic activity and engines of growth[39]. Food web research focuses on understanding the effects of removing species, such as through extinction[40], adding new species, such as through invasion[41], and consequences for systems stability. It also unveils how such interactions create systems-wide and coevolutionary effects across entire populations[42]. We take a similar approach to understanding the stability of mine-town systems in reaction to the proposed switching of energy systems.

In mine-town systems, mining projects are the main centres of economic activity and growth. The Food Web Model avoids positioning specific population nodes at the centre of analysis by focusing on interactions within the overall system. Primarily used in ecology science to study consumer-resource systems[43,44], this model provides a more accurate guide for characterising the complex linkages and dependencies embedded in mine-town systems. The Food Web Model bears some resemblance to the energy-shed concept, where scholars approach the spatial relationships in energy systems as analogous to a watershed[45,46].

By drawing on the Food Web Model, our approach characterises the 'feeding' interactions within mine-town systems. The relationship between mining projects and towns becomes a hierarchical network of supply interactions between: (i) mining projects and towns, and (ii) towns and cities of different size and importance. These mine-town interactions propagate socio-economic assets and functions throughout the system. As with trophic levels in a food web, mining projects represent producers and towns represent consumers. Supply interactions across the system are bi-directional, providing functions that feed population demand and a population that in turn feeds the functions.

In this way, we define a mine-town system as a network of socio-economic connections between resource inventories and town systems. A mining project, as part of the resource inventory, provides functions (e.g. employment) for dependent towns, while towns provide the demographic content (e.g. employees) to feed the project. These socio-economic interactions take place via direct (first 1° level) and indirect (second 2° and third 3° level) linkages (Fig. 1). Smaller towns supply the population to feed functions (e.g. services) to the larger town, and so forth. The number of functions and the size of population to feed (or to be fed by) corresponds with the hierarchical structure of the mine-town system. Larger towns, such as cities, carry greater cumulative importance in the system through their wider sphere of influence than smaller towns. Larger towns attract population from a broader geographic catchment given the functions they offer. Smaller towns are more spatially proximite to mining projects and feel the immediate effects of change. Figure 1a depicts a mine-town system. Figure 1b reflects our definition using an example of the

Canadian Sudbury mine-town system (listed in Table 1). In this system, 715 mining projects from the resource inventory provide socio-economic functions via direct and indirect interactions to 18 rural towns, 7 urban towns and the regional city of Sudbury. The mine-town systems approach enables the examination of system functionality and viability under different scenarios—such as climate change.

Table 1 shows the top ten mine-town systems in our dataset. These systems are concentrated in Canada and Australia with one region in the United States. In depicting the spatial organisation of mine-town systems, we utilise the closest geographical straight-line distance from towns to mining projects and between towns at different levels as a proxy for identifying socio-economic interactions across the system. We apply a two-step procedure. For 1° direct interaction between a mining project and the closest settlement, a threshold of 200 km travel distance was set as a maximal commuting distance from a settlement to a mining project[47,48]. For 2° and 3° indirect interactions, we depict the closest higher-level settlement. Data on global settlements were sourced from GHS Settlement Model layers (GHS-SMOD)[49] and Global Rural-Urban Mapping Project: Settlement Points[50]. GHS-SMOD enables the hierarchical classification of towns. Because real settlements are not points but larger areas, these data were smoothed onto a 10 km square grid, a scale that better represents the typical physical extent of settlements on a global scale (as used by Corbane et al.[51]). Each grid cell with a settlement was then classified according to the hierarchical level of that settlement (i.e. rural town, urban town, regional city), with priority assigned to the higher-level settlement.

The S&P database is global in coverage and relies on public disclosures by mining companies, noting that its completeness levels are not consistent globally, due to varying disclosure standards in different jurisdictions and across companies. In general, Russia, China and India are likely to be underrepresented, whereas project information for Australia, Canada and the United States are considered more reliable. A detailed description of the data and design of the mine-town systems is provided in Methods and in Source Data.

## Impacts of shifting commodity markets

The Intergovernmental Panel on Climate Change[52] estimates that in 1.5 °C pathways renewables are projected to supply 70–85% of electricity in 2050, while the use of coal shows a steep reduction to close to 0% of electricity. This projection is based on rapid and far-reaching transitions in energy systems. Reflecting this projection, our study elaborates on two working assumptions supported by recent developments in research and industry that we refer to as phase-out and phase-in assumptions. First, under the phase-out assumption, the pressure to discontinue coal will increase in the next three decades[53]. Whether due to policy, pricing, market changes or resource depletion, the working assumption is that nearly every open mine will eventually close. This may lead to comprehensive mine closure or abandonment as an eventuality for all coal projects in production. Second, under the phase-in assumption, a low-carbon future will be metal intensive as clean energy technologies need more metals than fossil fuel-based technologies. Demand for energy transition metals is predicted to grow rapidly, leading to a larger material footprint worldwide[54]. Consequently, our working assumption is that a greater share of pre-operational ETM projects may become operational under acute pressure to balance out global demand[55]. In addition, the International Energy Agency[56] reported that there is too little ETM mining investment to satisfy the energy transition-driven demand for ETMs. This is on top of pressures to phase-out coal.

The transition will therefore impact resource inventories and consequently all mine-town systems (Fig. 2). To illustrate the potential impacts of shifting commodity markets on the stability of mine-town systems, this study builds and then analyses three global Resource Inventories (based on data from S&P Global[57]). Inventory I includes coal mining projects. Inventory II includes energy transition metal

(ETM) projects where the calculation of Reserves and Resources are known. This inventory represents the notionally available stock of ETMs. In this study, ETMs refer to 17 minerals and metals required to support the phase-in of low-carbon energy technologies. They include aluminium, chromium, cobalt, copper, graphite, indium, iron, lead, lithium, manganese, molybdenum, neodymium, nickel, silver, titanium, vanadium and zinc[55]. Large quantities of iron, aluminium, copper and nickel are also used in non-energy applications. Inventory III includes all known ETM projects across all project development stages. It includes the projects identified in Inventory II plus projects where no Reserves and Resources were disclosed. The distinction between Inventory II and III is that ETM projects with declared Reserves and Resources (Inventory II) are more likely to receive attention from investors—a feature that can be taken as increasing the likelihood of project development compared to a project with no declared Reserves and Resources (Inventory III). This is not absolute, however, as we describe in the Limitations. Inventory III includes both the notionally available and the undefined stock of ETMs over which technology markets and states will seek to have influence. Figure 3 shows mining projects for each of the resource inventories according to the project's development stage.

Mine-town systems represent socio-economic connections between resource inventories and demographic systems. To assess the geographical extent of the impact on mine-town systems in response to the coal phase-out and ETMs phase-in assumptions, we analysed 35,831 mining projects in the three Resource Inventories (based on data from S&P Global[57]), and 246,921 settlements (represented by 10 km square settlement grids) of different sizes and levels in town hierarchies as part of demographic systems (data from CIESIN[50] and Florczyk et al.[49]). We used a sequential proximity analysis to establish linkages between mining projects and towns and identify mine-town systems. Mining projects and their location form the base elements of the mine-town systems, with settlement size and the closest proximity additional features. Establishing linkages between projects in Resource Inventories I, II and III with different types of human settlements enables the analysis of mine-town systems on a global scale.

We identified 95.8% of global mining projects across the three Resource Inventories (a total of 34,319) and 7.6% of global settlements (a total of 18,669) as being a part of global mine-town systems, including 3023 regional cities, 4325 urban towns and 11,321 rural towns. From Inventory I, 5184 coal projects are connected to settlements in a mine-town system (15.1% of mine-town systems globally). In terms of project life cycle, 51% of these coal projects are operating, 27% are in the pre-operational stage and 13% are closed. From Inventory II, 5330 ETM projects with known Reserves and Resources are connected to settlements in a mine-town system (34.4% of mine-town systems globally). 65% of these projects are in the pre-operational stage, 32% are operating and 4% are closed. From Inventory III, 16,877 projects are connected to settlements in a mine-town system (nearly 50% of mine-town systems globally). Seventy-four percent of these projects are in the pre-operational stage, 19% are operating and 4% are closed. Exemplars of mine-town systems in Australia, the state of Queensland and the Mackay mine-town system are presented in Fig. 4.

Potential effects of the transition on the architecture of mine-town systems for mine closure and mine opening are presented in Fig. 2. To analyse these effects, we included operational and closed coal projects for the phase-out assumption. For the phase-in assumption, we worked with pre-operational and operational ETM projects. The most pronounced effects in these town-mine systems occur in the nearest proximity between mine and town. After this, the mine-town effects are cumulative and indirect. The direct effects increase with decreasing size of the settlement due to lower levels of economic diversity. This also relates to geographical proximity—smaller towns are effectively satellites connected to a central demographic hub

**Table 1 | The top mine-town systems globally by number of mining projects**

| No. | Regional city | Country | # Mining projects | Commodities (# Mining projects[a]) | # Rural towns (Minimal estimated population) | # Urban towns (Minimal estimated population) |
|---|---|---|---|---|---|---|
| 1 | Prince George | Canada | 1051 | Gold (829), Copper (483), Silver (395), Zinc (202), Lead (171), Molybdenum (145), Coal (47), Nickel (25), Platinum (23), Palladium (23), Cobalt (18), Rhodium (17), Lanthanides (16), Tungsten (13) | 24 (>12,000) | 10 (>50,010) |
| 2 | Perth—Armadale | Australia | 984 | Gold (745), Nickel (294), Copper (187), Cobalt (111), Silver (73), Zinc (70), U3O8 (47), Iron Ore (51), U3O8 (47), Platinum (46), Lead (42), Palladium (41), Lithium (36), Tantalum (22) | 8 (>4000) | 3 (>15,003) |
| 3 | Sudbury | Canada | 715 | Gold (482), Copper (209), Nickel (156), Platinum (106), Palladium (100), Silver (85), Zinc (69), Cobalt (58), Rhodium (51), Diamonds (49), Lead (45), Graphite (22), U3O8 (21), Lanthanides (11) | 18 (>9000) | 7 (>35,007) |
| 4 | North Bay | Canada | 711 | Gold (544), Copper (211), Silver (162), Zinc (132), Nickel (65), Platinum (44), Palladium (42), Diamonds (39), Cobalt (37), Lead (37), Rhodium (21), Lithium (19), Lanthanides (18), Molybdenum (14), U3O8 (14) | 30 (>15,000) | 6 (>30,006) |
| 5 | Saguenay | Canada | 626 | Gold (279), Copper (195), Silver (93), Iron Ore (92), Nickel (85), Zinc (67), Diamonds (64), U3O8 (56), Palladium (50), Platinum (48), Cobalt (35), Lead (35), Lanthanides (30), Titanium (27), Rhodium (24), Magnetite (23), Phosphate (19), Lithium (18), Graphite (16), Vanadium (15), Molybdenum (14) | 45 (>22,500) | 14 (>70,014) |
| 6 | Sparks | USA | 566 | Gold (521), Silver (227), Copper (56), Lead (23), Zinc (21), Molybdenum (16), Lithium (11) | 11 (>5500) | 3 (>15,003) |
| 7 | Perth—Joondalup | Australia | 554 | Iron Ore (233), Gold (195), Copper (136), Silver (72), Lead (65), Zinc (59), Nickel (39), Lithium (35), U3O8 (33), Tantalum (30), Manganese (25), Cobalt (22), Tin (18), Platinum (16), Diamonds (15), Ilmenite (15), Palladium (15), Heavy Mineral Sands (14), Zircon (14), Rutile (14) | 18 (>9000) | 4 (>20,004) |
| 8 | Thunder Bay | Canada | 508 | Gold (352), Copper (161), Silver (100), Platinum (93), Palladium (92), Nickel (87), Zinc (74), Lead (42), Rhodium (42), Molybdenum (23), U3O8 (20), Cobalt (19), Iron Ore (19), Lithium (18), Tantalum (14), Diamonds (13), Lanthanides (12) | 13 (>6500) | 2 (>10,002) |
| 9 | Darwin | Australia | 455 | Gold (172), Copper (138), U3O8 (99), Diamonds (85), Zinc (74), Lead (69), Silver (65), Nickel (50), Iron Ore (36), Platinum (25), Cobalt (25), Lanthanides (22), Palladium (21), Phosphate (18), Manganese (15), Bismuth (12) | 23 (>11,500) | 6 (>30,006) |
| 10 | Edmonton | Canada | 447 | U3O8 (180), Diamonds (122), Gold (99), Copper (47), Silver (36), Nickel (27), Zinc (20), Platinum (15), Lead (15), Palladium (13), Cobalt (12), Coal (11), Lanthanides (11), Lithium (11) | 34 (>17,000) | 12 (>60,012) |

[a]Listed commodities mined/ to be mined in more than 10 projects per mine-town system.

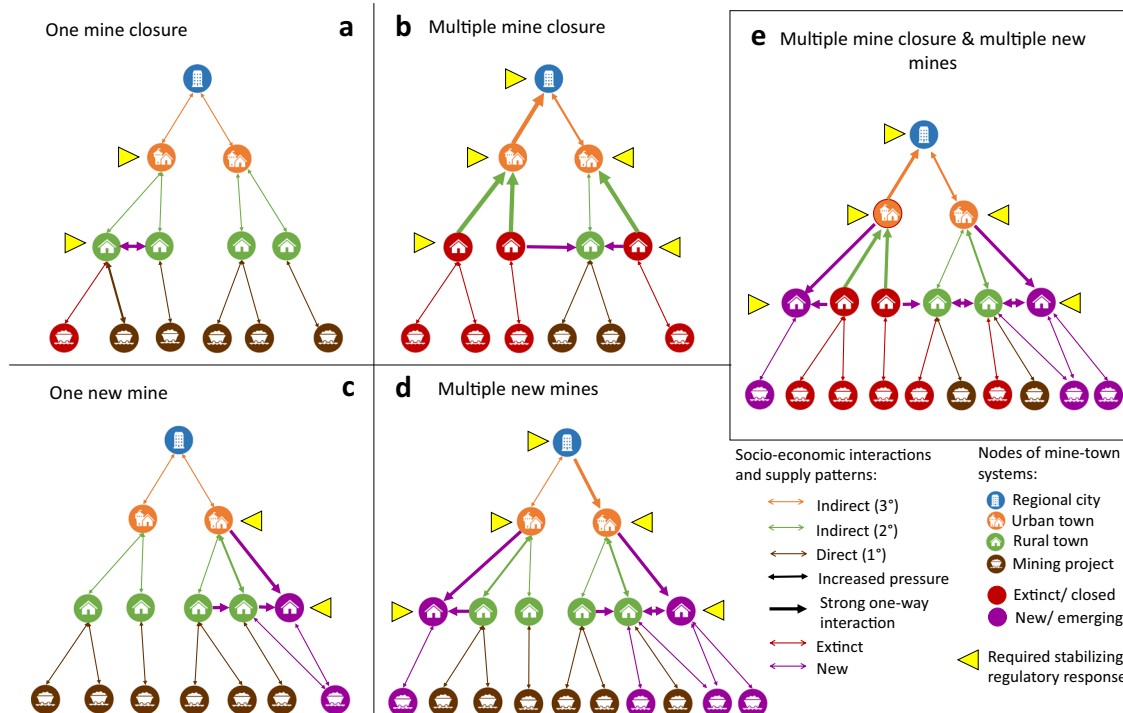

**Fig. 2 | Impacts of mine closure and mine opening suggested across the mine-town systems. a** A closure of a single project has a direct impact on an associated rural town, which leads to increased pressure on other rural towns in the system. Overexpansion of a rural town requires a stabilising regulatory response at the rural and urban towns levels. **b** Multiple mine closures drive out-migration from associated rural towns to urban towns, putting pressure on new host locations. A stabilising regulatory response is required at all town levels. **c** A new mining project is potentially followed by the establishment of a new associated rural town. This triggers changes in socio-economic interactions between neighbouring rural and urban towns towards the newly developed nodes in the system. **d** Multiple new projects trigger similar impacts on the system as **b** with downward pressure on socio-economic flows in the system's hierarchy. **e** Accumulation of both mine closure and mine opening doubles the pressures on the regulatory response to stabilise the system with overexpansion occurring in rural and urban towns and regional cities.

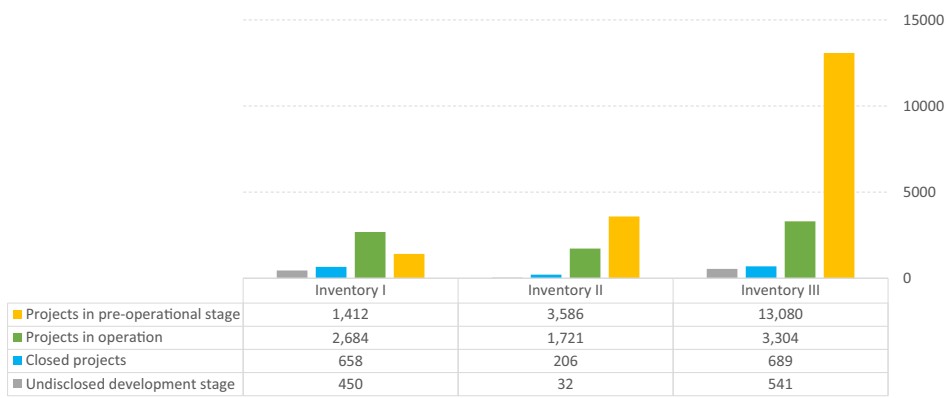

| | Inventory I | Inventory II | Inventory III |
|---|---|---|---|
| Projects in pre-operational stage | 1,412 | 3,586 | 13,080 |
| Projects in operation | 2,684 | 1,721 | 3,304 |
| Closed projects | 658 | 206 | 689 |
| Undisclosed development stage | 450 | 32 | 541 |

**Fig. 3 | Inventories according to the development stage of mining projects.** Inventory I includes coal mining projects. Inventory II includes energy transition metal projects where the calculation of Reserves and Resources are known. Inventory III includes the projects identified in Inventory II plus projects where no Reserves and Resources were disclosed.

(regional cities). When these satellites are closer to mines, and further from cities, the low economic diversity drives higher rates of economic dependency on mining projects and creates the potential for more adverse consequences at closure or abandonment of the asset. Supplementary Information 1 presents examples of three mine-town systems, where the potential impacts of the transition are described, considering the socio-economic conditions of the populations in these mine-town systems. While Example 1 (EMalahleni, South Africa) and Example 3 (Mackay, Australia) show the mine-town systems with prevailing coal phase-out transition, communities in Example 2 (Salta, Argentina) may face a rapid ETM phase-in transition.

Our analysis of Inventory I shows that ~10% (1780) of settlements in the mine-town system may face direct impacts of mine closure or abandonment. These settlements include 185 regional cities, 238 urban towns and 1357 rural towns with a total estimated population of at least 11,119,000 people that are located in the nearest proximity to coal projects. Most rural settlements (93%) do not have a future mining alternative. In addition, 879 settlements may be impacted by indirect and cumulative impacts as they are in socio-economic interaction with coal projects from a higher hierarchical position in the mine-town system. Of these, 479 are regional cities with a total population of at least 23,950,479 people and 400 are urban towns (estimated

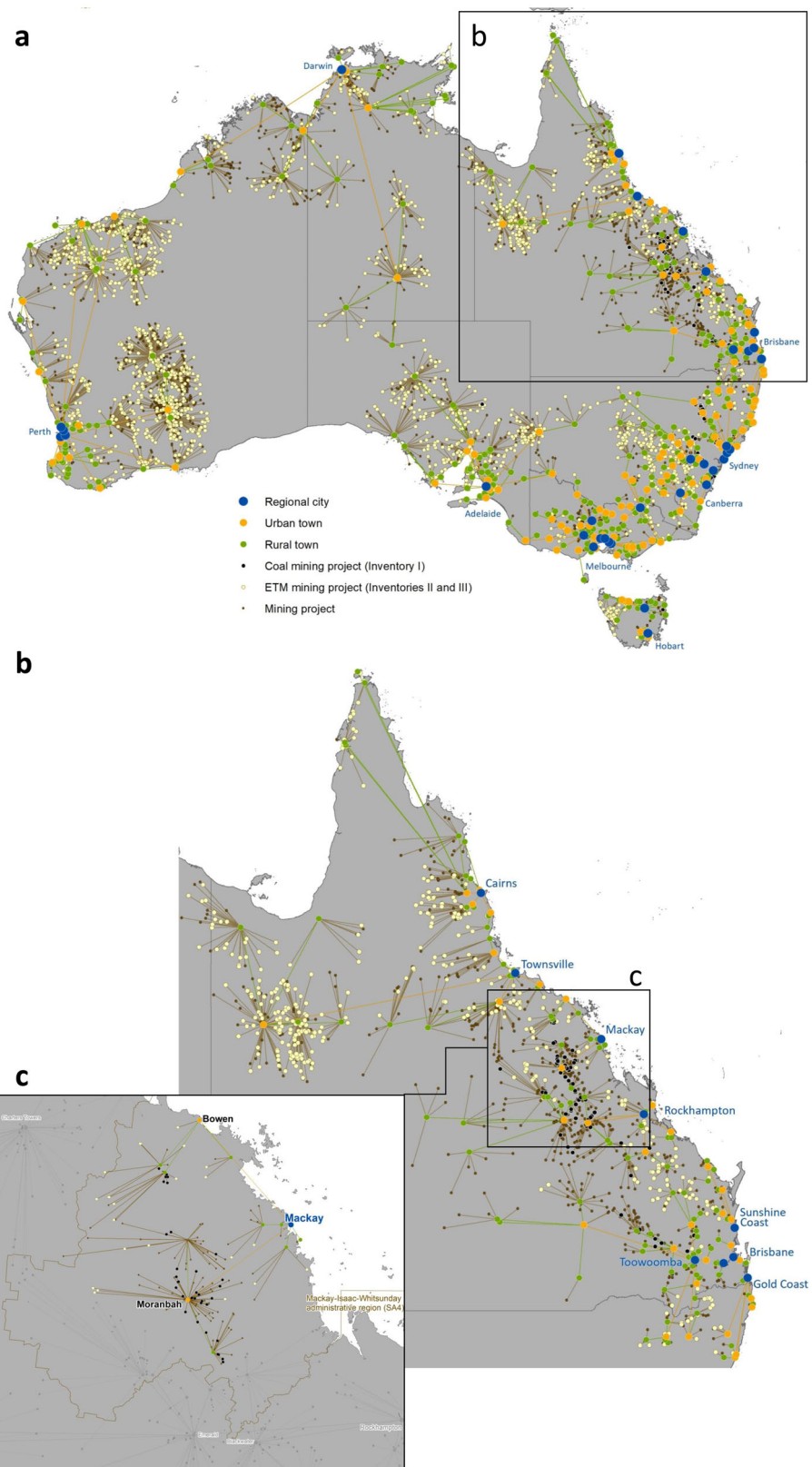

**Fig. 4 | Mine-town systems in Australia. a** National overview, **b** detailed geography of the state of Queensland, **c** Mackay mine-town system.

population of at least 2,000,400 people). If mine closures are poorly planned, and accumulate within mine-town systems, unintended social, economic and environmental consequences are likely to ensue[18,58]. While some specialised coal mines may remain open for high value material, such as metallurgical coal for steel production or coal

reserves for backup power, the broad demographic effects relating to Inventory I remain the same. Unplanned mine closure can trigger job losses, out-migration, disintegration of social structures, decline of infrastructure and the cessation of essential services. It can also induce increases in resentment, community opposition and even populism[59].

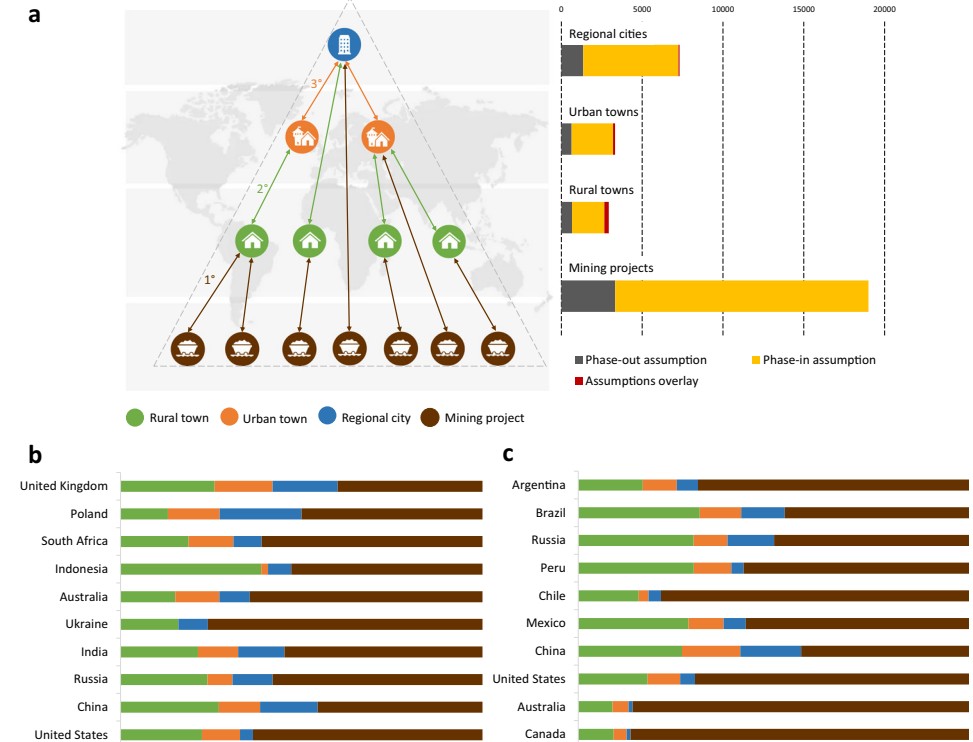

**Fig. 5 | Global exposure to the coal phase-out and ETM phase-in assumptions carried in the three resource inventories. a** Number of regional cities, urban and rural towns and mining projects in global mine-town systems. **b** Top 10 countries with the highest exposure to the coal phase-out assumption, i.e. countries with the largest number of settlements and operational and closed coal mines that are linked inside mine-town systems. **c** Top 10 countries with the highest exposure to the ETM phase-in assumption, i.e. countries with the largest number of settlements and pre-operational ETM projects that are part of the mine-town systems.

Entire towns can be abandoned, with cascading impacts on larger towns in the mine-town system through corresponding in-migration. When mines that feed towns close in the absence of system-level coordination, social and economic decline is a known outcome[60]. Even when mine closure is planned, the task of diversifying economies and revitalising industrial landscapes can be a lengthy, fraught and expensive process[1,16,61,62]. Any attempt to manage the social, economic and environmental impacts of mine closure requires institutional capacity[10] and a proactive approach[63], with additional pressure in regions facing multiple mine closures, at scale.

As the demand for new energy technologies grows, so too will demand for ETMs. We suggest that all ETM projects in Inventories II and III would need to be operational to meet the global demand. Our modelling shows that the 3216 settlements (17%) directly linked to Inventory II may be directly impacted by these developments, including 111 regional cities, 375 urban towns and 2730 rural towns (estimated population of at least 37,950,000 people). Another 2179 settlements (12%) are indirectly linked to Inventory II, including 1119 regional cities and 1060 urban towns (at least 108,951,000 people), and may face indirect and cumulative effects associated with the growing demand for ETMs.

Supplementary Information 2 presents the estimated global population in the mine-town systems linked to Inventory II for the 17 studied ETMs.

Analysing Inventory III, our modelling shows that 6917 (37%) settlements in mine-town systems are directly linked to this resource inventory, with another 3531 (16%) settlements linked indirectly. Two hundred sixty-seven regional cities, 762 urban towns and 5888 rural towns (a minimum of 80,890,000 people) may be directly impacted with another 1731 regional cities and 1800 urban towns (a minimum of 176,551,000 people) indirectly connected. Sixty-one percent of the rural towns linked with Inventory III are not linked to any existing mining projects and are set to become part of a mine-town system as projects move into operation. These towns may be drawn into mining-dominated economies as feeders. Mining has the potential to generate economic benefits for nearby towns and their rural-urban systems. Accruing these benefits would require government investment in social services and infrastructure to support burgeoning populations including, in some cases, new towns to service new mining projects. Each of these new mines will, at some stage, need to close—leading the mine-town systems back towards the socio-economic consequences of mine closure or abandonment.

Supplementary Information 3 provides a global overview of coal and ETM projects and their associated settlements in the mine-town systems affected by the phase-out and phase-in assumptions. Supplementary Information 4 shows global heat maps of the mine-town systems that may be affected by the coal phase-out and ETM phase-in assumptions.

Our results show that only 1% of settlements in mine-town systems will be affected if the transition effects of both Inventories I and III occur concurrently (see Fig. 5a). This reflects the fact that thermal coal and energy transition metals projects tend not to be co-located. Only 139 settlements will experience the direct impacts of this commodity switch with another 327 (2%) settlements experiencing indirect impacts. These towns may be best positioned to prosper through rapid workforce re-deployment from a coal economy to an energy transition metals economy, although they would face intense pressure to coordinate the switch, and the mine closure liabilities that would eventually follow.

## Remoteness in mine-town systems

Studies about the viability of remote towns emphasise inequalities between remote towns and regional centres. The trend toward increasing inequality between rural and regional locations is a global

phenomenon[64–66]. Peripheralization describes a process by which rural communities have environmental harms imposed on them by urban communities, and how such processes create unfair economic and geographic conditions alongside environmental and cultural inequalities[67].

Research shows that many remote towns face depressed economic conditions, rising costs of service delivery and infrastructure maintenance and sustained out-migration of educated youth. Most governments seem unable to systematically address these structural issues through regulatory means and remote forms of governance[68]. When these challenges accumulate, they result in a general decline in the quality of life for people who remain living in remote locations. In this paper, we associate the impacts of the phase-out and phase-in assumptions with remoteness within these systems.

We analysed the remoteness of rural towns in mine-town systems using the 'dispersed' and 'hinterland' categories of the Urban-Rural Catchment Areas (URCA) dataset[64]. The dataset provides comprehensive global gridded data on catchment areas of different population sizes. In URCA, remote towns in these categories are located more than 3 h from an urban agglomeration of at least 20,000 people. We identified 1169 remote rural towns in mine-town systems directly impacted by the assumptions carried in the three resource inventories (Fig. 6). Only 79 remote rural towns directly linked to Inventory I may be affected by the complete eventual phase-out of coal. Another 1090 remote rural towns in the mine-town systems with Inventory III, of which 477 remote rural towns linked to Inventory II, may be directly affected by bringing all available ETM projects into production. From those, 23 remote rural towns are exposed to both the closure and ramp-up propositions implied in Inventories I and III. The long-lasting social and economic viability of mining towns facing co-occurrence may be compromised if mining is added or removed from their economic structure[68] in the absence of a reconfiguration and reorientation of the systems that support these remote regions. Our study develops a novel approach to analysing the geographical complexity of the energy transition by connecting the location and type of resource with the dependencies and contingencies attached to town systems. Through a novel mine-town systems approach, we assess the dynamics of the socio-economic interactions between three resource inventories and mining populations under two working assumptions supported by recent developments in the energy transition agenda: first, the pressure to discontinue coal leads to total coal phase-out; second, that a metal intensive low-carbon future creates acute demand to bring pre-operational energy transition metal projects into operation.

We find, as Table 2 summarises, that the potential impacts of the coal phase-out and ETMs phase-in may not be experienced uniformly across the resource inventories. There is a severe spatial concentration in three countries—Australia, Canada and the United States. Moreover, there are notable differences across the inventories. As Table 2 reveals, the bulk of the mine-town systems set to be affected by the phase-out of coal is in the United States. This finding underscores the need for further action on Just Transition and efforts to protect workers and communities dependent on coal revenue[69,70]. Indonesia, China, Australia and Russia are also on this list, and could also benefit from action on Just Transitions[71]. Policymakers in these locations may seek to hedge these risks by better designing impact-benefit agreements to protect workers and communities or stronger protections related to impact monitoring or revenue distribution. The Table also underscores possible gaps in data within our assessment as there may be more mine-town systems than we present in our results (particularly in Asia, Russia, Africa and South America). This points toward the need for local and national governments to collect more reliable data on resource inventories and insist on more rigorous reporting in Africa, South America and Asia, including China.

Some of the world's major cities were once mining towns. These towns grew exponentially before populations stabilised and governments became involved in planned services and infrastructure. The phase-out of coal in the mine-town systems of Inventory I shows the population distribution across rural mining towns and the importance of stabilising the economic and service structure of larger towns as smaller towns fall into decline. The phase-in of energy transition metals highlights the level of investment required, not only to bring new energy transition metals projects through to production, but also the demands that will be placed on the service infrastructure needed to support new mine-town systems. As our data reveals, 93% of rural towns (1262) in coal-associated mine-town systems do not have a future mining alternative. At the same time, most of these towns are located less than a 3-hour commute from an urban town or city. These town locations may be especially vulnerable to abandonment given the lower likelihood of economic alternatives. The 1262 towns with populations between 500 and 5000 each represent a total estimated population of 631,000–6,310,000 people presently residing in town systems without an economic alternative to mining. Closing mines that feed these rural towns may alter the requirements for services and employment to the larger towns and cities in the systems. Residents of these rural towns may migrate to the larger towns, which may in turn cause increased pressure on housing, related infrastructure and services. Alternatively, residents may commute from rural towns to larger centres, putting pressure on transportation systems and infrastructure and increasing carbon emissions.

Consequently, our research validates concerns about the impacts of ramping down coal and ramping up energy transition metals on town systems in the absence of improved data, analysis and planning. The finding suggests, as summarised in Table 3, that the potential impacts on populations associated with ramping up all available energy transition metals (Inventory III), or even only those with known Reserves and Resources (Inventory II), could intersect with significantly larger populations than the ramping down of coal. Within our global sample of mine-town systems, we identified four times more settlements associated with developing all available energy transition metals projects from Inventory III and twice the number of settlements associated with developing all known ETM projects from Inventory II than with existing coal mines in Inventory I (53% vs. 29% vs. 15% of all settlements in mine-town systems). For instance, 3592 rural towns in the energy transition metals mine-town systems linking Inventory III are not located near an existing mine. These towns were previously not exposed to mining and may therefore be drawn into the disruptive boom and bust cycles of mining-dominated economies. These rural towns could experience indirect and cumulative mining-induced impacts across their town systems. This represents an estimated total population of 1,796,000–17,960,000 people who may face systems disruption in their administrative, social and economic systems if all available energy transition metal projects are brought into operation. Moreover, a further 1090 remote rural towns will be directly proximate to resource development complexes if all available energy transition metals projects are brought to market. 79 remote rural towns may be directly affected by acute coal mine closure. Fourteen (14) times the number of remote rural towns intersect with, and could be negatively impacted by, ETM inventories compared with all coal projects.

## Discussion

The overall population effects are difficult to distil given the high number of inter-connections across these town systems. Conservatively, we estimate, and Table 3 summarises, that a minimum of 33.5 million people could be affected by the phase-out of coal and another 70 million affected if only energy transition metals projects with known Reserves and Resources become operational, and 115.7 million people affected if all available energy transition metals projects become operational.

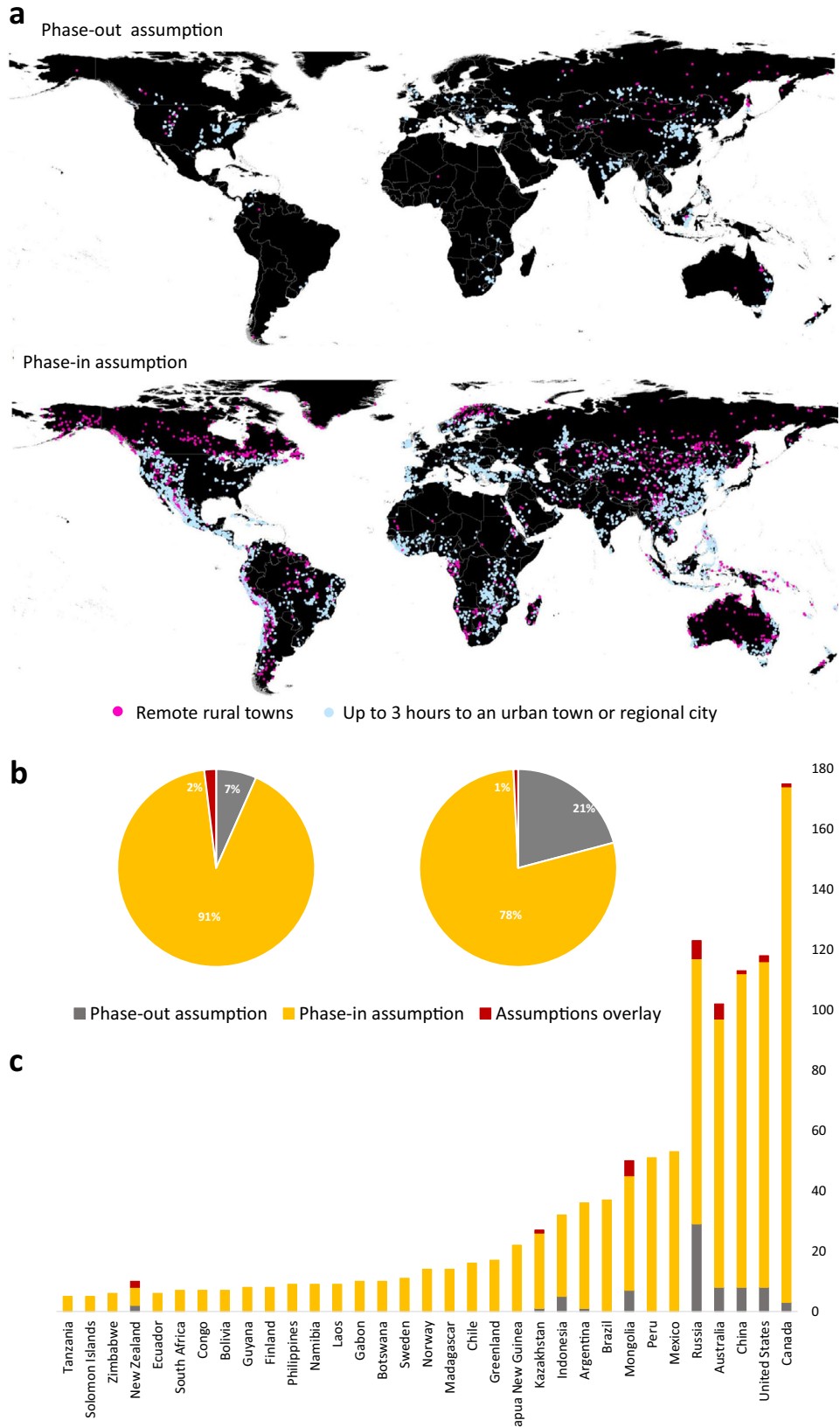

**Fig. 6 | The impact of phase-out and phase-in assumptions on remote rural towns and mine-town systems. a** Geography of the remote rural towns (in pink) and rural towns reachable in up to 3 h (in blue), which are part of the global mine-town systems impacted by coal phase-out and ETM phase-in assumptions carried in the three resource inventories. **b** Global exposure of the remote rural towns to the assumptions and their overlay. **c** National exposure of the remote rural towns to the assumptions and their overlay.

**Table 2 | The most vulnerable mine-town communities to total coal phase-out in Inventory I (a) contrasted with those vulnerable to the partial (Inventory II) or total phase-in (Inventory III) of energy transition metal mining projects (b)**

(a) Top 10 mine-town systems by number of coal mining projects under coal phase-out

| No. | Regional city | Country | # Mining projects | # Coal (Inventory I) | % Coal to phase-out | # Rural towns | # Urban towns | Population affected | % National population |
|---|---|---|---|---|---|---|---|---|---|
| 1 | Huntington | USA | 329 | 329 | 60.2 | 44 | 5 | >47,005 | 0.014 |
| 2 | Charleston | USA | 225 | 225 | 65.3 | 34 | 5 | >42,005 | 0.013 |
| 3 | Pittsburgh | USA | 182 | 182 | 67.0 | 92 | 39 | >241,039 | 0.073 |
| 4 | Evansville | USA | 80 | 79 | 78.8 | 33 | 13 | >81,513 | 0.025 |
| 5 | Johnson | USA | 88 | 87 | 63.6 | 14 | 4 | >27,004 | 0.008 |
| 6 | Knoxville | USA | 90 | 82 | 52.2 | 17 | 9 | >53,509 | 0.016 |
| 7 | Birmingham | USA | 54 | 54 | 72.2 | 19 | 5 | >34,505 | 0.010 |
| 8 | Shenmu | China | 42 | 42 | 92.9 | 24 | 3 | >27,003 | 0.002 |
| 9 | Newcastle | Australia | 83 | 60 | 43.4 | 19 | 12 | >69,512 | 0.270 |
| 9 | Banjarbaru | Indonesia | 64 | 59 | 56.3 | 47 | 2 | >33,502 | 0.002 |
| 9 | Roanoke | USA | 45 | 44 | 80.0 | 13 | 4 | >26,504 | 0.008 |
| 10 | Mackay | Australia | 138 | 106 | 24.7 | 9 | 2 | >14,502 | 0.056 |
| 11 | Kiselyovsk | Russia | 39 | 39 | 87.2 | 5 | 0 | >2500 | 0.002 |

(b) Top 10 mine-town systems by number of energy transition metals mining projects to phase-in, including the mine-town systems impacted by partial development of known ETMs (Inventory II), and by bringing all available ETM projects into operation (Inventory III)

| No. | Regional city | Country | # Mining projects | # ETMs (Inventory II) | # ETMs (Inventory III) | % ETMs to phase-in (Inv. II&III) | # Rural towns | # Urban towns | Population affected | % National population |
|---|---|---|---|---|---|---|---|---|---|---|
| 1 | Prince George | Canada | 1051 | 554 | 677 | 62.7 | 24 | 10 | >62,010 | 0.162 |
| 2 | Perth—Armadale | Australia | 984 | 825 | 984 | 43.8 | 8 | 3 | >19,003 | 0.074 |
| 3 | Perth—Joondalup | Australia | 554 | 406 | 554 | 71.8 | 18 | 4 | >29,004 | 0.113 |
| 4 | Saguenay | Canada | 626 | 529 | 626 | 53.5 | 45 | 14 | >92,514 | 0.242 |
| 5 | North Bay | Canada | 711 | 625 | 711 | 41.6 | 30 | 6 | >45,006 | 0.118 |
| 6 | Sudbury | Canada | 715 | 656 | 715 | 39.2 | 18 | 7 | >44,007 | 0.115 |
| 7 | Adelaide | Australia | 366 | 305 | 366 | 72.7 | 36 | 11 | >73,011 | 0.284 |
| 8 | Sparks | USA | 566 | 492 | 566 | 46.1 | 11 | 3 | >20,503 | 0.006 |
| 9 | Kelowna | Canada | 326 | 274 | 326 | 75.8 | 23 | 8 | >51,508 | 0.135 |
| 10 | Thunder Bay | Canada | 508 | 453 | 508 | 48.2 | 13 | 2 | >16,502 | 0.043 |

**Table 3 | Estimated global population thresholds of the mine-town systems in Inventory I impacted by total coal phase-out, partial development of known ETMs (Inventory II) and bringing all available ETM projects into operation (Inventory III)**

| Settlement level | Population thresholds | Mine-town systems linked to Inventory I | | Mine-town systems linked to Inventory II | | Mine-town systems linked to Inventory III | |
|---|---|---|---|---|---|---|---|
| | | Direct impact | Indirect impact | Direct impact | Indirect impact | Direct impact | Indirect impact |
| Rural town | 500–5000 | 678,500–6,785,000 | 2,395,479–23,950,000 | 1,365,000–13,650,000 | 5,301,060–53,000,000 | 2,944,000–29,440,000 | 9,001,800–90,000,000 |
| Urban town | 5001–50,000 | 1,190,238–11,900,000 | | 1,875,375–18,750,000 | | 3,810,762–38,100,000 | |
| Regional city | >50,000 | >9,250,185 | >20,000,400 | >5,550,111 | >55,951,119 | >13,350,267 | >86,551,731 |
| Population affected | | >11,118,923 | >22,395,879 | >8,790,486 | >61,252,179 | >20,105,029 | >95,553,531 |

While the timing associated with the coal phase-out and energy transition metals phase-in is unknown, the mine-town systems approach outlines the extent of demographic, infrastructure and economic planning attached to wholesale propositions about energy markets involving coal and its alternatives. Our findings reveal potential negative synergies between larger and smaller towns. For example, closing smaller towns further down in the network could have major indirect and cumulative consequences in larger town systems that would otherwise be considered stable. In addition, the mine-town systems approach provides new clarity on the regional-scale constraints to phase-out for A1 'recalcitrant' countries (i.e. USA, Canada and Australia) identified by Svobodova et al.[13]. These are high emitting nations that have the economic means to transition, but not the political will.

In addition to identifying the sobering potential for disruption in future clean energy transitions, our study also points the way towards research gaps and future work. More granular, robust regional-scale data on the socio-economic conditions of populations living and working within these mine-town systems could inform future policy-making. More granular data would enable policymakers to address, for example, the distribution of impacts by age, race, gender, socio-economic status and other intersectional issues. This data could come from population surveys, focus groups, or qualitative interviews, or use ethnographic and anthropological techniques to capture the "lived experiences" of these communities, similar to other work on cobalt mining in the Congo[72], e-waste scrapyard workers in Ghana[73] or shale gas extraction in the UK[74].

Such data can help inform the development of fit-for-purpose policies, regulations, plans and investment frameworks to support these transitions and will require coordination and adaptation within and across states and industries. Economic planning and development policies need to adopt a broader territorial perspective that takes account of the interlinkages between settlements of different sizes in hierarchical mine-town systems. For example, our mine-town systems modelling identified employment as a key feature and the supply of employees as a systems linkage. This linkage has implications for rural-urban road, rail and other transport networks, utilities such as water and energy supply in different locations, and access to services such as health, education and banking and finance. Current planning and policy frameworks target established administrative boundaries and economic activities, such as farming and agriculture within defined regions[64]. This conventional perspective can overlook complex linkages across a broader territorial domain, limiting our ability to identify and counter the pressures that will be placed on mine-town systems as demand for coal declines, and the extraction of energy transition metals intensifies.

Ultimately, without more adaptive policy frameworks, effective governance and attention to justice issues, the global low-carbon transition threatens to bring severe economic hardship to thousands of mine-town systems and millions of households.

## Methods

### Empirical approach

The mine-town systems approach assesses the linkages between resource inventories and human settlements. Every mining project located at a commuting distance from a settlement is part of a hierarchical mine-town system. The growth dynamics of the system depend, to some extent, on mining projects in the system. Drawing on work on the functional hierarchy and self-organisation of complex systems[75,76], we used the principles underpinning the Food Web Model[43] as the basis of our approach. Christaller's[77] concept of 'maximum distance', the straight-line geodesic distance between projects and settlements and between settlements, serves as a proxy for socio-economic interactions across the mine-town system. This is the main criteria for identifying projects and settlements in the system. The analytical premise is that mines have common characteristics and their

linkages to settlements are independent of the commodity being extracted. The time scale assumptions carried in the three resource inventories described in the study reflect carbon reduction targets set for 2050.

Using geographical proximity as a proxy for identifying socio-economic interactions between demographic systems and resource inventories is supported by the theory of spatial economics, where spatial dependence is one of the main determinants of economic growth[78]. As towns compete for residents, the commuting distance to mining employment becomes an important source of a population centre's competitive advantage. Close geographical proximity to a workforce similarly supports the economics of a project by reducing costs[79,80].

### Modelling of mine-town systems

The mine-town system definition comprises three hierarchical levels of settlements: rural town, urban town and regional city. The higher a settlement sits in the system hierarchy, the larger its population size and number of socio-economic functions. As socio-economic diversity increases, economic dependence on mining decreases[81]. At this scale, however, the effects become cumulative and multi-directional. The classification of settlements follows the Global Human Settlement Layer (GHSL) analytics framework[49]:

(i) Rural towns (populations between 500 and 5000 inhabitants) represent small towns and large villages. These towns provide accommodation for mine workers and low-order services such as elementary schools, post offices, health clinics and local markets.

(ii) Urban towns (populations between 5001 and 50,000 inhabitants) represent regional towns that fulfil functions as higher-order service access centres and transport nodes for the most proximate rural towns.

(iii) Regional cities (population of more than 50,000 inhabitants) serve as hubs for larger regional areas. These larger centres provide access to high-order goods, facilities and services that are not available in smaller urban and rural towns, such as higher education, and certain personal and professional services. As centres of the world economic system, these cities have political influence, and often host national or local government agencies. Their national and international transport links facilitate access to a large base of suppliers and customers.

To identify mine-town systems on an international scale, we generated a global set of existing and emerging mining projects. A worldwide sample of 35,891 mining projects was sourced from the S&P Global Market Intelligence database[57], a commercial database built on public disclosures from mining companies (data are current as of January 2021). We used the most complete project-level records in the S&P, including development stage, commodity and geographical coordinates. Development stages were grouped into three categories: (i) pre-operational, (ii) operational and (iii) closed. Projects with missing records on development stage and coordinates were excluded from the analysis, leaving a sample of 35,319 mining projects.

To generate the hierarchically classified layer of global settlements, we synthesised two data sources: GHS Settlement Model layers (GHS-SMOD)[49] and the Global Rural-Urban Mapping Project: Settlement Points, Revision 01 (GRUMP)[50]. GHS-SMOD is a raster grid at 1 km$^2$ with each cell classified as either a rural area, town and suburb, or city. This classification is achieved via cell clustering based on estimated population size, population density and built-up area density. The GRUMP dataset includes the localisation of 70,629 geo-referenced settlements with populations greater than 5000 persons. The main advantage of the GRUMP dataset is that it uses direct census population data, rather than being an estimation, as is the case for GSH-SMOD. The urban towns and regional cities were therefore sourced from GRUMP, while rural towns (cells coded 13)—the smallest scale of

human settlements in our mine-town systems, were sourced from GSH-SMOD. To reduce the potential for error in geo-locating settlements, we used a 10 km square grid, a distance chosen to better represent the typical physical extent of settlements. Using the 10 km settlement grid cells also greatly improved the speed and control of the computational analysis. Each grid cell containing a settlement was classified according to the hierarchical level of that settlement, with priority assigned to the higher-level settlement (e.g. if the cell contained both a rural and an urban town, it got the value of the urban town). In the subsequent analysis of mine-town systems, each cell was represented by its centre point (centroid). This procedure generated 213,489 cells classified as rural towns, 24,421 cells classified as urban towns and 9011 classified as regional cities.

By applying a sequential proximity analysis, we modelled mine-town systems analysing socio-economic interactions between settlements and mining projects based on three specific resource inventories. The settlement hierarchy determined the point of reference for a given level in the mine-town systems. First, we identified the closest settlement to a mining project (marked as 1° direct socio-economic interaction; see Fig. 1). Where rural and urban towns were identified in this interaction, we continued with the second and third sequence of the proximity analysis to map 2° and 3° indirect interactions until we identified a regional city as the highest hierarchical level of the mine-town system. To avoid cross-national linkages inside the systems, the analysis was limited to national borders. In other words, no cross-border linkages or flows were considered. For each 1° direct socio-economic interaction, we set a threshold of 200 km travel distance as a maximal commuting distance from a settlement to a mining project. Guided by Öhman and Lindgren[82] (for Sweden), Maoh and Tang[48] and Axisa et al.[47] (for Canada), we assumed that a maximum of 200 km straight-line distance is equal to ~2–4 h of travel time necessary for workers to reach mining projects regularly using ground transportation (depending on local conditions). The criterion of maximal travel distance excluded fly-in-fly-out commuting schemes as our aim was to capture functioning mine-town systems with regular socio-economic flows between towns and project locations. From the sample of 35,319 projects, 1000 projects were located farther than 200 km from a settlement, leaving a final sample of 34,319 projects being part of the mine-town systems.

Detailed characteristics of identified global mine-town systems are in the Source Data file.

### Remoteness in mine-town systems

To analyse the remoteness of the rural towns directly linked to coal and energy transition metals projects in the mine-town systems under the phase-out and phase-in assumptions, we used a dataset of Urban-Rural Catchment Areas (URCA)[64]. URCA is a comprehensive and detailed gridded dataset mapping catchment areas of different population sizes and providing their spatial representation across global demographic systems. For all rural towns in the mine-town systems identified as impacted by the assumptions, using ArcGIS we extracted URCA raster values (an attribute 'URCAmapord' coded 1-30) at the locations of the towns and identified the level of remoteness for each rural town in the mine-town systems. The URCA categories of Dispersed towns and Hinterlands (coded 29 or 30 in 'URCAmapord') were considered as remote. The rural towns located under these categories were considered as remote in our analysis.

### Limitations

(a) The functional scope of the mine-town systems
This study relies on data listed in the S&P Global Market Intelligence database[57]. While the S&P database is one of the most comprehensive sources of mining project data, it builds on public disclosures by mining companies and is therefore

incomplete. Data quality and availability in the database are not consistent across the globe because of varying disclosure standards in different jurisdictions and companies. These different standards affect the coverage, completeness and reliability of disclosures. Informal mining activities such as artisanal and small-scale mining are typically not well captured. The number of projects recorded in the S&P Capital IQ Pro database are based on public filings and third party resources used by S&P Global Market Intelligence. Data coverage for certain jurisdictions may be limited due to lack of data transparency whereas jurisdictions with stricter public disclosure guidelines may provide better coverage, extensiveness and consistency. (personal comm. Alexis De Leon). As such, there are many more mine-town systems than those we identify in this study.

We use reserves and resources (R&R) as one measure for demonstrating that a project is progressing along a development trajectory. Predicting the likelihood of project development involves many other dynamic factors. These include orebody characteristics, project design, availability of infrastructure, corporate risk appetite and financial position, in addition to pricing, and market demand. Furthermore, individual S&P records cannot be analysed temporally. Records provide a point-in-time statement of a project's status. Where disclosed, it is possible to extract brief notes of events and the time of their occurrence. The unevenness of corporate disclosures means it is not possible to devise a sufficiently standardised or sufficiently complete set of records to demonstrate the progress/regress of project development pathways, at scale.

Available global datasets also limit the functional scope of the mine-town systems. For instance, sub-national data for health, education, or sector-based economic performance does not exist at global scale. While small town municipalities, or even mines within these mine-town systems, may collect data on services or economic performance, this information is (i) rarely publicly available and (ii) therefore not available for the purposes of determining levels of usage (of services) or the local economic impact of introducing or withdrawing individual industries. The complex orebody research by Valenta et al.[83] problematizes the assumption that project development can be achieved by overcoming discrete technical challenges. Large-scale copper projects such as Pebble in Alaska, Resolution Copper in the United States and Tampakan in the Philippines show that constraints to development extend beyond whether R&R have been defined.

Limitations also relate to the datasets of global settlements GHS-SMOD and GRUMP. This data is better for larger settlements than for smaller settlements. This is problematic because the most direct effects within the mine-town systems occur at the closest proximity to the mine, and the majority of mining projects globally are located further away from the largest settlement category in our approach. Another limitation of the global population datasets is inaccurate geo-location in the case of vector data, and scale dependence and uncertainty in population density estimate at fine scales in the case of raster data[84]. The overall population effects of the energy transition were estimated using the latest data in GHS-SMOD and GRUMP. Considering projections of world population growth[85], future demographic effects may be greater than we identify in this study.

Our model uses sequential proximity analysis by calculating the geodesic straight-line distance from each mining project to the closest settlement, and from this settlement to the closest higher-level settlement, et cetera. The geodesic straight-line distance does not consider accessibility to transport infrastructure[64,66,86]. While geodesic straight-line distances serve as a proxy for travel time in research about service planning[87,88], and are considered equivalent to travel distance[87,89], accessibility and transport data would add precision. The proximity analysis is designed to avoid cross-border linkages in the mine-town systems, which may represent a limitation in countries where the socio-economic flows between resource inventories and town systems may be cross-national. As such, the identified mine-town systems may contain more mining projects and settlements than presented in our study. Including the transport data and cross-national connections in our analysis is an avenue for future research. As the settlement hierarchy is defined in the source data itself, and the interactions between projects and settlements and between settlements are defined by closest-distance links, the analysis involves only one free parameter, the grid size. This represents a real-world quantity (the size of settlements) and thus cannot vary greatly from the adopted value.

Despite these limitations, our work brings a theoretical understanding closer to real-world conditions than previous studies.

(b)    Limitations external to the mine-town systems

It is not possible to predict the future price of ETM commodities. There are simply too many varying contingent factors that determine future pricing[90–92]. Advancements in technology and knowledge of material substitution will drive, to some considerable extent, the bundle of commodities required for renewable energy technologies and this in turn will influence the pressure applied to ETMs projects to supply those materials. The effect of political, environmental, or financial market events impacts heavily on both supply and demand[23,24]. The extent and predictability of these events were not estimated in this study.

## Software
Microsoft Excel software was used for collection of mining project data. All spatial data analyses were conducted using ArcGIS (version 10.7.1)[93].

## Reporting summary
Further information on research design is available in the Nature Portfolio Reporting Summary linked to this article.

## Data availability
The data generated in this study are provided in the Source Data file. The dataset sourced from S&P Global Market Intelligence database, that support the findings of this study can be obtained from the corresponding author upon reasonable request. Source links of other publicly available datasets used for this study (GHS-SMOD, GRUMP and URCA) are provided in References. Source data are provided with this paper.

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

## Author contributions

Conceptualisation: K.S., J.R.O., M.S., É.L., V.M., D.K. and B.K.S.; Methodology: K.S., D.K., M.S., V.M., É.L. and J.R.O.; Investigation: K.S., V.M. and M.S.; Writing—original draft: K.S., D.K., J.R.O., B.K.S., É.L., V.M. and M.S.; Writing—review and editing: K.S., D.K., J.R.O., B.K.S., É.L., V.M. and M.S.; Visualisation: K.S., V.M. and M.S.

## Competing interests

The authors declare no competing interests.
