## [Peer Review File · Nature Communications]

Decarbonization, population disruption and resource inventories in the global energy transitionREVIEWER COMMENTS

Reviewer #1 (Remarks to the Author):

The authors offer a novel approach to understanding the connection between mining projects and geographic (i.e., urban, suburban, and rural) dimensions. They cleverly apply the 'Food Web Model' to center mining projects rather than cities in their study. This is a valuable approach to consider; however, the results are limited. I recommend a list of minor and major revisions below.

Minor:

While the authors explain their approach to conceptualizing 'mine-town systems,' I remain confused as to the specifics of how a 'mine-town system' is defined. I would recommend choosing one of the mine-town systems listed in table 1 as an example and taking the reader through the steps of how that system was defined.

The first paragraph (lines 125-130) should be moved to the top of the preceding section. It feels plopped in at the wrong place.

Starting on line 135, the authors make several claims about coal production. I recommend a citation to support their assertion.

I recommend putting the information about the projects in each of the three inventories into a table. It is a lot to digest in the text and difficult to determine how the inventories compare. The table could distinguish what type of projects are included in each inventory then the total number of projects, those that are in pre-operational stage, those in operation, those closed, and those in development, etc.

Figure 5(a) is very small and difficult to read.

In the conclusion, the authors offer raw numbers of people impacted by mining transitions. I recommend including proportions as well as the raw numbers to give the reader a better idea of scale.

Major:

The results of this piece are interesting because they offer estimates on the number and type of geographic areas (i.e., cities, towns, settlements, etc.) – and thereby number of people – potentially impacted by coal phase-out and ETM phase-in. However, I think these results fall a bit flat and should be explored more deeply. Just indicating the number of people impacted seems insufficient for the caliber required by Nature Communications. Using this data or other data sources can the authors answer more nuanced questions about how these mining transitions will impact people living in these mine-town systems? One idea is to consider the socioeconomic conditions of the people that live within these mine-town systems (e.g., income levels now versus what they might be if the mine were to close; impacts on jobs and, importantly, types of jobs). I understand this is likely impossible for all systems, but maybe it is worthwhile to present this larger, novel framework and then pick one system to explore in much deeper detail as a case study. This would allow the audience to understand why this new framework is necessary and how it can be used in future research. I imagine this may be a big lift, but I think it is worthwhile for the authors to consider.

Reviewer #2 (Remarks to the Author):

Thank you for inviting me to review this manuscript, which seems very interesting/insightful. Aside from well-written, I find the manuscript potentially impactful, with strong policy- and science-relevant takeaways on a knowledge gap that is well-established in the Introduction. I do have a list of both major and minor comments/concerns; however, overall, I believe that revisions in response to my comments should be fairly manageable.

Oddly, I think the title and abstract do not do justice to the research:

- The title seems overly ambitious: disruptions between resource inventories and mining populations are not really assessed; the highlighted message should be about the potentially disrupting coal phase-out and the parallel phase-in of materials that are considered critical in the global energy transition, but the entailed disruptions are not wholly assessed. Perhaps "Identifying potential disruptions" rather than assessing them?
- The abstract reads slightly detached from the main text. What range of effects of switching from coal to clean energy technologies are modelled? Also, the subsequent sentence reads off (lines 15-17). Furthermore, findings do not necessarily suggest that ETM phase-in would be more disruptive to demographic systems than coal phase-out, but perhaps potentially disruptive to larger demographic systems/populations. There are many takeaways underplayed/missing in the abstract, so perhaps the abstract could be rewritten.

Introduction: Between lines 68 and 69, we miss a clear statement of what the paper is about. Paragraph in lines 69-78 establishes that it addresses the gaps highlighted above, but there is no clear statement of what the study does. Also, "the human dimension" (line 72) is not really assessed, but part of that.

Supplementary Data 1: Why are there mining project IDs missing from this dataset? Do missing ID values imply anything (e.g., ID 2703, first missing instance)? The ratio of identified vs. max ID (34,319/35,731) is very close to the 95.8% figure mentioned in Line 185 of the manuscript, but this is not confidently the same figure and it is not entirely clear why not all mining projects were identified: is the remainder not part of a mine-town system (i.e., remote mining projects not sufficiently adjacent to settlements) or not about the metals covered in Inventories I-III (coal + ETMs)? . Also, I understand that each row corresponds to a unique mining project; each mining project appears to have only up to one 1-level direct as well as 2-level and 3-level indirect interactions, which is counterintuitive and in conflict with Figure 1.

Lines 197-198: Authors probably mean Supplementary Data 2, not 3. Also, in that, are we to assume that mine-town systems affected by phase-in include pre-operational ETM systems and mine-town systems affected by phase-out include pre-operational and operational coal systems?

Lines 221-222: The authors suggest that all ETM projects must be operational to meet global demand; without really contesting this, is this suggestion based on anything else than an assumption, or is there any sufficiency/availability estimate behind this?

Lines 221-227: Much like before, does this paragraph refer to pre-operational mining projects? If so, should it be spelled out?

Lines 242-243: I believe this is poorly phrased: perhaps the intended message is that 1% of settlements in mine-town systems will be affected by both assumed processes of the net-zero transition? The way I read the original phrasing means that 1% of the settlements will not be affected if one of the two assumptions stand, but the idea is that this 1% will be affected if either of the two assumptions stand (and even more so if both stand – or less so, in case of targeted policy interventions for workforce shifts).

Line 261: Scenarios I and II possibly refer to coal phase-out and ETM phase-in, respectively. The authors should revise/rephrase, as there currently is no definition of scenarios in the study (and rightly so).

Lines 266-269: Perhaps authors should add "remote" to the 79, 1090, 477 figures (i.e., "79 remote rural towns"), as it currently reads odd, invalidating all previous findings.

Lines 269-270: "will experience the affected of both Inventories I and III", please rephrase to make sense (although it's clear what the intended message is).

Lines 282-291: I am not confident this part shouldn't be instead in the previous section. Also, in lines 290-291, perhaps it would be best to note that this is "based on the available dataset".

Lines 304-305: Saying that most of these towns are located less than a 3-hour commute from an urban town or city is not very insightful/interesting. I believe it is more interesting to turn the argument around, specifying that a small part of these rural towns are also remote, meaning that without mining alternatives nor any broader economic activity diversity they are potentially facing abandonment.

Lines 313-321: Again, this to me seems newly introduced, so why isn't this discussion part of the Results and Discussion section?

Supplementary Data 3: I understand the need to have these huge ranges throughout the text, as these draw from the definition of rural (500-5,000) and urban (5,001-50,000) towns. However, contrary to Population affected in Table 3 (which seems correct), Population impacted in Supplementary Data 3 seems exaggerated to the upper bounds: authors seem to have added the upper population bounds of rural and urban towns (wrong) to the minimum bounds of regional cities (correct). If I am reading this correctly, for example, Silver should say Population impacted: 4,477,263 (direct) & 38,021,376 (indirect), instead of 20,920,053 and 68,800,692 respectively. This considerably tones down the findings about the numbers of populations to be impacted by ETM phase-in, per metal and totally.

Lines 385-386: Do these time scale assumptions imply a concrete timeline for project phase-out/-in, or is it simply 2050? This is a general remark/question: what is the considered timeline and which socioeconomic trajectory have the authors followed for population trends? If we expect immediate phase-out and -in, which is not the case, then population numbers can be based on today's data, otherwise there must be assumptions about how these numbers will evolve, e.g., by 2050.

Lines 674-678: There is something off here; the Svobodova et al. ref is also below (line 679), and the Sun et al. ref is "hidden" in this bullet. I believe Svobodova et al. here must be deleted from this bullet.

Figure 1: Much like in the caption, I found that the main text could/should benefit from a definition of a mine-town system. Also, in the figure, shouldn't there be assumed interactions among same-level nodes that may be adjacent (e.g., a rural town with another rural town)?

Figure 2: I suspect impacts are not modelled but rather theoretically assumed; or is there anything quantifiable in this figure (looking at strong interactions, for example)?

Figure 3: breaking the caption on (b) and (c) down (i.e., "assumptions" aside), in essence they refer to countries with largest populations (or, according to lines 754-755, perhaps largest numbers of settlements) being directly or indirectly linked to pre-operational and operational coal mining projects (b) or to pre-operational mining projects of any ETM. However, I assume that resource potential and/or level of dependence play no role in these rankings. Lines 754-755 define the axis as the number of settlements and mining projects in mine-town systems; I am not confident this is clear to me. These two figures (b and c) definitely need better explanation in the caption.

Broader methodological questions:

If I understand correctly, the authors used empirical data from GRUMP to identify settlements—except for the smallest (rural) towns, for which predictive data from GSH-SMOD—and applied a 10km^2 grid ($10\text{km} \times 10\text{km}$) to locally assess population numbers (or, perhaps, other settlements) and characterising each grid as one of the three settlement levels.

a. On this, authors refer to "point centre", not sure what this entails, but my understand is that the higher level is prioritised (i.e., if grid population $>50\text{k}$, then regional city; otherwise, if $>5\text{k}$, then urban town; otherwise, if >500 , then rural town). The authors are kindly requested to confirm and/or clarify in the text.

Then, to determine direct socioeconomic interactions, the authors set a threshold of 200km travel distance around each settlement, assuming this is <2hr commuting distance.

b. Not sure this is entirely accurate; 200km distance is hardly achievable in less than 2 hours, in most parts of the world and depending on the quality of the transport network and available means of transport. This is especially true if the 200km distance is geodesic straight line, which is the case here. This also applies to indirect interactions, as my understanding is that 2-level indirect flows essentially imply direct flows among towns/cities, one of which is directly linked to a mining project. My main concern here is that the mine-town system network, as built in this study, may be denser than in reality.

Remoteness of rural towns was, finally, assessed using a different dataset (URCA).

c. How are three-hour distances identified here? Is it based on the 100km/hour assumption and geodesic straight-lines (i.e., 300km), or does this come straight from URCA for the two categories (dispersed, hinterland)? Also, my possibly trivial question is whether authors could have assessed remoteness based on their existing mine-town system mapping (i.e., rural towns without 2-/3-level interactions).

Finally, this all comes down to a question on the quantitative output of the analysis:

d. The entire study has a purely geographic/population (density) perspective that may not give the full picture; could results be highly sensitive on the size of the grid, the node-to-node distance threshold, etc.? Perhaps different specs/thresholds may have yielded different results, qualitatively in the same scale for the population impacted but not so much for the number/type of settlements of interest (i.e., within a mining project-encompassing cell) and the density of the mine-town systems? This comment is not intended to contest the core finding of the study, of course: coal phase-out and critical metals phase-in are assumed to be an integral part of any Paris Agreement-compliant transition, but we tend to neglect their possible socioeconomic implications for massive parts of the global population outside the typically modelled economic dimensions (X USD investments, Y job losses/gains, Z GDP shifts, etc.).

This long list of questions/remarks and recommendation for major revisions should be interpreted as an intention to help enhance the quality, transparency, and readability of the paper, and do justice to the novelties presented. It is a well-elaborated study, with significant work underpinning it, informative visualisations and strong takeaways.

A. N.

Reviewer #3 (Remarks to the Author):

Overall:

Very interesting and important topic that is assessed in an innovative way. Covering both the decline of coal and the increase in mining for ETMs is a fair approach. The review of towns and cities as an ecosystem works well and shows how these communities are linked. Paper would greatly benefit from more clear definition of terms as well as more clarity about metrics of impacts, both positive and negative. The database gap that misses information on mines in Asia (China, India), Russia, and Africa, needs to be better explained and managed, perhaps with a change in the focus of the paper. Specific comments below.

Specific Comments:

Line 12-23, Abstract: While explained in the paper, the use of the terms "disrupt" and "sensitive to" in the abstract are ambiguous about whether it is positive disruption or negative. Change can often be good although it be viewed quite differently by various stakeholders. Suggest authors revise the abstract to be more specific about the types of impacts.

Line 81+: The definition of a mine-town system is important and should be summarized in the text as early as possible. The closest to a clear definition is in the caption of Figure 1 but is still missing key information. The details are well described in lines 387+ in the Methods that the journal format puts at the end of the article, but leaves the reader unclear what they are when reading the main text. Additionally, the term "town" is a smaller population as defined in Supplemental Data 3 but unclear here that the "systems" part of the term "mine-town systems" includes the cities. Concept Is fine but emphasizes the need for clearer definitions in the paper up front, since they are scattered across the text, captions, methods, and supplemental data.

Line 120: The lack of identified mine-town systems in China, as well as Russia and India, in the top 20 seems like a serious data gap. Also, much of the concern of ETM mining has been for communities in Africa and South America and they do not seem to be on the list. This seems like enough of a gap that perhaps the paper should be recontextualized as a study of North America, Australia, and perhaps Europe. However, later in the paper, Figure 3 shows Russia, China, Africa, etc. So the geographic scope of this study needs to be clarified or made consistent.

Line 136: Some specialized coal mines may remain open for high value material such as metallurgical coal for steel production or coal reserves for backup power. So I think the authors can say nearly every open coal mine or functionally every coal mine will close but maybe not all. Results will be the same.

Lines 210-220: Can this be balanced with any positive effects of mine closures, such as towns able to diversify and transform their economies? Many former coal mining towns in the western U.S. grew and benefited after their mines closed, although it can take decades.

Lines 237-239: Similar to previous comment, what are the negatives to communities of starting or expanding a mine, in addition to the eventual closure?

Lines 290-291: Will the bulk of mine-town systems affected by the energy transition really be in Canada or Australia, or is that an artefact of incomplete data sets? Vast amounts of mined materials that go into renewable energy come from China, as well as Africa and South America, so it is difficult to believe Canada will be most impacted. The data gaps are mentioned in lines 293-295 but they are significant enough that lines 290-291 seem like an overstatement that could be taken out of context.

Line 345: Unclear what 'recalcitrant' means here, as this is the first and only time this term is used in this paper.

Link 437-438. 200 km/2 hours seems excessive for commute time, especially when global medians are about 30 minutes. If 2 hours is viewed as a maximum it seems that the likelihood of a community being a mine-town would decline as distance increases past the median. Or perhaps this does not mean daily commutes? Authors need to define the basis for this assumption and how the distances were selected.

Line 458+: Are artisanal mines included? These have some of the highest community impacts. While it may not be possible to include them, they should be mentioned in the limitations.

Supplemental Data 2 maps. For large countries, such as Russia, China, and the U.S., the subnational data may be more important than the number of projects and settlements per country. A heat map of the cities would be more representative than country scale.

Reviewer Response table for NCOMMS-22-19417

Reviewer 1

#	Reviewer Comment	Authors' response
1.1	The authors offer a novel approach to understanding the connection between mining projects and geographic (i.e., urban, suburban, and rural) dimensions. They cleverly apply the 'Food Web Model' to center mining projects rather than cities in their study. This is a valuable approach to consider; however, the results are limited. I recommend a list of minor and major revisions below.	Thank you. (No response required, addressed below.)
1.2	Minor: While the authors explain their approach to conceptualizing 'mine-town systems,' I remain confused as to the specifics of how a 'mine-town system' is defined. I would recommend choosing one of the mine-town systems listed in table 1 as an example and taking the reader through the steps of how that system was defined.	We clarify the definition in the text of the manuscript and add Figure 1b with a real-world example as suggested.
1.3	Minor: The first paragraph (lines 125-130) should be moved to the top of the preceding section. It feels plopped in at the wrong place.	Thank you for this helpful suggestion. We have modified the text as recommended.
1.4	Minor: Starting on line 135, the authors make several claims about coal production. I recommend a citation to support their assertion.	A citation is now provided.
1.5	Minor: I recommend putting the information about the projects in each of the three inventories into a table. It is a lot to digest in the text and difficult to determine how the inventories compare. The table could distinguish what type of projects are included in each inventory then the total number of projects, those that are in pre-operational stage, those in operation, those closed, and those in development, etc	A new graph is now available in Figure 2. This figure includes the information requested by the Reviewer.
1.6	Minor: Figure 5(a) is very small and difficult to read.	We have enlarged this Figure slightly for ease of review however, we are mindful that this will likely be resolved through the production process.
1.7	Minor: In the conclusion, the authors offer raw numbers of people impacted by mining transitions. I recommend including proportions as well as the raw numbers to give the reader a better idea of scale.	Thank you for this suggestion. In Table 2, we now present the percentage of the national population that may be affected by the transitions for all top 10 mine-town systems to provide an indication of scale.

1.8	Major: The results of this piece are interesting because they offer estimates on the number and type of geographic areas (i.e., cities, towns, settlements, etc.) – and thereby number of people – potentially impacted by coal phase-out and ETM phase-in. However, I think these results fall a bit flat and should be explored more deeply. Just indicating the number of people impacted seems insufficient for the caliber required by Nature Communications. Using this data or other data sources can the authors answer more nuanced questions about how these mining transitions will impact people living in these mine-town systems? One idea is to consider the socioeconomic conditions of the people that live within these mine-town systems (e.g., income levels now versus what they might be if the mine were to close; impacts on jobs and, importantly, types of jobs). I understand this is likely impossible for all systems, but maybe it is worthwhile to present this larger, novel framework and then pick one system to explore in much deeper detail as a case study. This would allow the audience to understand why this new framework is necessary and how it can be used in future research. I imagine this may be a big lift, but I think it is worthwhile for the authors to consider.	Very good point about the need for better socio-demographic data. Because in many cases this data is non-existent, we have now drafted this new paragraph in the Conclusion: “In addition to identifying the sobering potential for disruption in future clean energy transitions, our study also points the way towards research gaps and future work. More granular, robust data on the socioeconomic conditions of the people that live within these mine-town systems could better inform future policymaking. This data could come from community surveys, focus groups, or qualitative interviews, or use ethnographic and anthropological techniques to capture the “lived experiences” of these communities, similar to other work on cobalt mining in the Congo (Sovacool 2019a), e-waste scrapyards workers in Ghana (Sovacool 2019b), or shale gas extraction in the UK (Sovacool et al. 2020).” Regarding the suggested case study, we have added Supplementary Data 2 where we present 3 examples of the mine-town systems with details about the socio-economic consequences of the phase-in and phase-out transitions in these systems. The examples are selected to show the different potential impacts of the transitions in different countries. See Example 1 – EMalahleni mine-town system, South Africa; Example 2 – Salta mine-town system, Argentina; Example 3 – Mackay mine-town system, Australia.
------------	--	--

Reviewer 2

#	Reviewer Comment	Authors' response
2.1	Thank you for inviting me to review this manuscript, which seems very interesting/insightful. Aside from well-written, I find the manuscript potentially impactful, with strong policy- and science-relevant takeaways on a knowledge gap that is well-established in the Introduction. I do have a list of both major and minor comments/concerns; however, overall, I believe that revisions in response to my comments should be fairly manageable.	Thank you. We agree that the comments are manageable.
2.2	Oddly, I think the title and abstract do not do justice to the research: - The title seems overly ambitious: disruptions between resource inventories and mining populations are not really assessed; the highlighted message should be about the potentially disrupting coal phase-out and the parallel phase-in of materials that are considered critical in the global energy transition, but the entailed disruptions are not wholly assessed. Perhaps “Identifying potential disruptions” rather than assessing them?	The title is changed to ‘An approach for identifying disruptions between resource inventories and mining populations in the global energy transition’.
2.3	- The abstract reads slightly detached from the main text. What range of effects of switching from coal to clean energy technologies are modelled? Also, the subsequent sentence reads off (lines 15-17). Furthermore, findings do not necessarily suggest that ETM phase-in would be more disruptive to demographic systems than coal phase-out, but perhaps potentially disruptive to larger demographic systems/populations. There are many takeaways underplayed/missing in the abstract, so perhaps the abstract could be rewritten	The reviewer is calling for greater precision in the abstract. We have revised the abstract based on this feedback and the feedback of the other reviewers.
2.4	Introduction: Between lines 68 and 69, we miss a clear statement of what the paper is about. Paragraph in lines 69-78 establishes that it addresses the gaps highlighted above, but there is no clear statement of what the study does. Also, “the human dimension” (line 72) is not really assessed, but part of that.	This has been amended to make the link between inventories and phase-out/phase-in thinking clearer. The term ‘human’ is replaced with ‘spatial’ and ‘demographic’ – linking directly the previous paragraph and the broader purpose of the study.
2.5	Supplementary Data 1: Why are there mining project IDs missing from this dataset? Do missing ID values imply anything (e.g., ID 2703, first missing instance)?	Thank you for pointing this out. The missing values are now corrected and the data is presented in complete series.
2.6	Also, I understand that each row corresponds to a unique mining project; each mining project appears to have only up to one 1-level direct as well as 2-level and 3-level indirect interactions, which is counterintuitive and in conflict with Figure 1.	Yes, this is correct. We have corrected the number of connections in Figure 1.

2.7	Supplementary Data 1: The ratio of identified vs. max ID (34,319/35,731) is very close to the 95.8% figure mentioned in Line 185 of the manuscript, but this is not confidently the same figure and it is not entirely clear why not all mining projects were identified: is the remainder not part of a mine-town system (i.e., remote mining projects not sufficiently adjacent to settlements) or not about the metals covered in Inventories I-III (coal + ETMs)?	The IDs are corrected in the Supplementary Data 1 – see our response to 2.5. The data cleaning procedure is clarified across the manuscript.
2.8	Lines 197-198: Authors probably mean Supplementary Data 2, not 3.	Yes. This is now corrected.
2.9	Lines 197-198: Also, in that, are we to assume that mine-town systems affected by phase-in include pre-operational ETM systems and mine-town systems affected by phase-out include pre-operational and operational coal systems?	Mine-town systems affected by phase-out (coal) include operational and closed projects. Mine-town systems affected by phase-in (ETMs) include pre-operational and operational projects. We have clarified this in the section Impacts of shifting commodity markets and in the methods.
2.10	Lines 221-222: The authors suggest that all ETM projects must be operational to meet global demand; without really contesting this, is this suggestion based on anything else than an assumption, or is there any sufficiency/availability estimate behind this?	Recent studies (e.g., International Energy Agency 2021 : p. 119) find that there is too little ETM mining investment to satisfy the energy transition-driven demand for ETMs. Known ETM projects, where investment has been committed (i.e. projects in Resource Inventory II and III), can only deliver a fraction of what is needed. We note however that there is uncertainty on the amount of recoverable ETMs in pre-operational-stage projects (particularly those in Resource Inventory III that do not have reserves and resources records in S&P). This is clarified in the methods.
2.11	Lines 221-227: Much like before, does this paragraph refer to pre-operational mining projects? If so, should it be spelled out?	We have clarified the logic behind our analyses across the manuscript.
2.12	Lines 242-243: I believe this is poorly phrased: perhaps the intended message is that 1% of settlements in mine-town systems will be affected by both assumed processes of the net-zero transition? The way I read the original phrasing means that 1% of the settlements will not be affected if one of the two assumptions stand, but the idea is that this 1% will be affected if either of the two assumptions stand (and even more so if both stand – or less so, in case of targeted policy interventions for workforce shifts).	We have improved the phrasing here. This is a minor editorial change showing that very few settlements would be co-located near to both a closing coal operation and an ETM project that was ramping up.

2.13	Line 261: Scenarios I and II possibly refer to coal phase-out and ETM phase-in, respectively. The authors should revise/rephrase, as there currently is no definition of scenarios in the study (and rightly so).	Yes, you are correct. This is now corrected to “the phase-out and phase-in assumptions”.
2.14	Lines 266-269: Perhaps authors should add “remote” to the 79, 1090, 477 figures (i.e., “79 remote rural towns”), as it currently reads odd, invalidating all previous findings.	Thank you for this. We have made this change.
2.15	Lines 269-270: “will experience the affected of both Inventories I and III”, please rephrase to make sense (although it’s clear what the intended message is).	Thank you – this has been changed to make explicit reference to the closure and ramp up propositions of the two inventories.
2.16	Lines 282-291: I am not confident this part shouldn’t be instead in the previous section. Also, in lines 290-291, perhaps it would be best to note that this is “based on the available dataset”.	This material has now been moved into the previous section.
2.17	Lines 304-305: Saying that most of these towns are located less than a 3-hour commute from an urban town or city is not very insightful/interesting. I believe it is more interesting to turn the argument around, specifying that a small part of these rural towns are also remote, meaning that without mining alternatives nor any broader economic activity diversity they are potentially facing abandonment.	Agreed. We added an additional sentence to bring this emphasis forward in the narrative. Thank you.
2.18	Lines 313-321: Again, this to me seems newly introduced, so why isn’t this discussion part of the Results and Discussion section?	This material now sits in the results and discussion section.
2.19	Supplementary Data 3: I understand the need to have these huge ranges throughout the text, as these draw from the definition of rural (500-5,000) and urban (5,001-50,000) towns. However, contrary to Population affected in Table 3 (which seems correct), Population impacted in Supplementary Data 3 seems exaggerated to the upper bounds: authors seem to have added the upper population bounds of rural and urban towns (wrong) to the minimum bounds of regional cities (correct). If I am reading this correctly, for example, Silver should say Population impacted: 4,477,263 (direct) & 38,021,376 (indirect), instead of 20,920,053 and 68,800,692 respectively. This considerably tones down the findings about the numbers of populations to be impacted by ETM phase-in, per metal and totally.	This is now corrected.
2.20	Lines 385-386: Do these time scale assumptions imply a concrete timeline for project phase-out/-in, or is it simply 2050?	2050 is the nominal target provided in the international declarations. Increasingly individual nation states are introducing policy measures to set out targets pre-2050; however the globally recognised declarations – far out into the future as they may happen to be – are the espoused targets.

2.21	Lines 385-386: This is a general remark/question: what is the considered timeline and which socioeconomic trajectory have the authors followed for population trends? If we expect immediate phase-out and -in, which is not the case, then population numbers can be based on today's data, otherwise there must be assumptions about how these numbers will evolve, e.g., by 2050.	No population trends have been modelled for this exercise – the primary projections relate to whether resource projects within the identified inventories will be activated or de-activated relative to the demographic systems in which they are located.
2.22	Lines 674-678: There is something off here; the Svobodova et al. ref is also below (line 679), and the Sun et al. ref is “hidden” in this bullet. I believe Svobodova et al. here must be deleted from this bullet.	Yes, this is correct. The references are now corrected.
2.23	Figure 1: Much like in the caption, I found that the main text could/should benefit from a definition of a mine-town system. Also, in the figure, shouldn't there be assumed interactions among same-level nodes that may be adjacent (e.g., a rural town with another rural town)?	See response to 1.2 above.
2.24	Figure 2: I suspect impacts are not modelled but rather theoretically assumed; or is there anything quantifiable in this figure (looking at strong interactions, for example)?	This is now corrected. We use “suggested” instead of “modelled”.
2.25	Figure 3: breaking the caption on (b) and (c) down (i.e., “assumptions” aside), in essence they refer to countries with largest populations (or, according to lines 754-755, perhaps largest numbers of settlements) being directly or indirectly linked to pre-operational and operational coal mining projects (b) or to pre-operational mining projects of any ETM. However, I assume that resource potential and/or level of dependence play no role in these rankings. Lines 754-755 define the axis as the number of settlements and mining projects in mine-town systems; I am not confident this is clear to me. These two figures (b and c) definitely need better explanation in the caption.	The caption of Figure 3 (now Figure 5) is now clarified in the caption, in particular (b) and (c).
2.26	Broader methodological questions: If I understand correctly, the authors used empirical data from GRUMP to identify settlements—except for the smallest (rural) towns, for which predictive data from GSH-SMOD—and applied a 10km ² grid (10km x 10km) to locally assess population numbers (or, perhaps, other settlements) and characterising each grid as one of the three settlement levels.	Yes, this is correct.
2.27	a. On this, authors refer to “point centre”, not sure what this entails, but my understand is that the higher level is prioritised (i.e., if grid population >50k, then regional city; otherwise, if >5k, then urban town; otherwise, if >500, then rural town). The authors are kindly requested to confirm and/or clarify in the text.	Yes, this is correct. We have clarified the approach and the terminology in the results and method sections.

2.28	Then, to determine direct socioeconomic interactions, the authors set a threshold of 200km travel distance around each settlement, assuming this is <2hr commuting distance.	We set a threshold of 200 km travel distance around each mining project in order to remove very distant mining projects that could possibly bias our analysis. Only the 1° interactions are limited to that distance. The projects farther than 200 km from any settlement in the dataset are excluded from mine-town system modelling. We have clarified this in the results and method sections.
2.29	b. Not sure this is entirely accurate; 200km distance is hardly achievable in less than 2 hours, in most parts of the world and depending on the quality of the transport network and available means of transport. This is especially true if the 200km distance is geodesic straight line, which is the case here. This also applies to indirect interactions, as my understanding is that 2-level indirect flows essentially imply direct flows among towns/cities, one of which is directly linked to a mining project. My main concern here is that the mine-town system network, as built in this study, may be denser than in reality.	We have explained our approach in the method section. Based on other studies, that we refer to in the text, we have refined that “a maximum of 200 km straight line distance is equal to approximately 2-4 hours of travel time.. (based on local conditions) ...”. We have further clarified the sequential proximity analysis used in modelling of mine-town systems. Regarding this concern, as we pointed out in the study limitations, there are probably more mine-town systems and more settlements included in mine-town systems than we identified.
2.30	Remoteness of rural towns was, finally, assessed using a different dataset (URCA).	Yes, this is correct.
2.31	c. How are three-hour distances identified here? Is it based on the 100km/hour assumption and geodesic straight-lines (i.e., 300km), or does this come straight from URCA for the two categories (dispersed, hinterland)?	The three-hour distance is taken from URCA. We have clarified this in the results and method sections.
2.32	Also, my possibly trivial question is whether authors could have assessed remoteness based on their existing mine-town system mapping (i.e., rural towns without 2-/3-level interactions).	Thank you for this comment. When developing the research methodology, we consider using GRUMP and GHS-SMOD settlement datasets. Because the URCA dataset is developed for this type of analysis and, at the same time, it is the most up-to-date and detailed dataset designed

		to identify catchment areas globally (including the remote ones), we decided to use URCA instead. Our analysis of remoteness in mine-town systems is based on a precise overlay of all rural settlements in mine-town systems with this dataset, using the extraction of URCA raster values in ArcGIS.
2.33	Finally, this all comes down to a question on the quantitative output of the analysis: d. The entire study has a purely geographic/population (density) perspective that may not give the full picture; could results be highly sensitive on the size of the grid, the node-to-node distance threshold, etc.? Perhaps different specs/thresholds may have yielded different results, qualitatively in the same scale for the population impacted but not so much for the number/type of settlements of interest (i.e., within a mining project-encompassing cell) and the density of the mine-town systems? This comment is not intended to contest the core finding of the study, of course: coal phase-out and critical metals phase-in are assumed to be an integral part of any Paris Agreement-compliant transition, but we tend to neglect their possible socioeconomic implications for massive parts of the global population outside the typically modelled economic dimensions (X USD investments, Y job losses/gains, Z GDP shifts, etc.).	The possible sensitivity to the grid cell and the node-to-node distance, as highlighted in the reviewer comment, is explained below and we also added more clarifications in the results and in the limitations. Because real settlements are not points, the settlement data are smoothed onto a 10 km square grid to better represent the typical physical extent of settlements. The grid size is selected based on the review of previous studies (e.g., Corbane et al. 2019) and pilot testing of the grid size. The 10x10 km grid cell is shown to be the best fit to the real-world extent of the settlements, covering proportionally small-sized rural towns and larger regional cities. This adjustment furthermore greatly improves the speed and control of the computational analysis. Each grid cell with a settlement is then classified according to the hierarchical level of that settlement with priority assigned to the higher-level settlement, meaning that if the grid cell contains both a rural and an urban town, it gets the value of the urban town. As this hierarchy is defined in the source settlement data itself, and the connections are defined by closest-distance links, this analysis involves only one free parameter, the grid size. This itself represents a

		real-world quantity - the size of settlements – and thus cannot vary greatly from the adopted value.
2.34	This long list of questions/remarks and recommendation for major revisions should be interpreted as an intention to help enhance the quality, transparency, and readability of the paper, and do justice to the novelties presented. It is a well-elaborated study, with significant work underpinning it, informative visualisations and strong takeaways.	Thank you again. The authors have accepted these recommendations in a positive and helpful tone, as the Reviewer indicates.

Reviewer 3

#	Reviewer Comment	Authors' response
3.1	Overall: Very interesting and important topic that is assessed in an innovative way. Covering both the decline of coal and the increase in mining for ETMs is a fair approach. The review of towns and cities as an ecosystem works well and shows how these communities are linked.	Thank you.
3.2	Paper would greatly benefit from more clear definition of terms as well as more clarity about metrics of impacts, both positive and negative.	This is agreed. Please refer to comments above and below for references to changes throughout the manuscript where these helpful suggestions have been incorporated.
3.3	The database gap that misses information on mines in Asia (China, India), Russia, and Africa, needs to be better explained and managed, perhaps with a change in the focus of the paper. Specific comments below.	Thank you for raising this concern, however the data used for this study is global, it is not biased north or south. We recognise that the original examples did not highlight the effects in less developed countries. This has been rectified by the inclusion of examples in Figures 5b, 6 and in the Supplementary Data 2, where we provide examples of three mine-town systems with details about the socio-economic consequences of the phase-in and phase-out transitions in these systems. The examples have been selected to show the different potential impacts of the transitions across north and south: EMalahleni, South Africa; Salta, Argentina; Mackay, Australia

3.4	Line 12-23, Abstract: While explained in the paper, the use of the terms “disrupt” and “sensitive to” in the abstract are ambiguous about whether it is positive disruption or negative. Change can often be good although it be viewed quite differently by various stakeholders. Suggest authors revise the abstract to be more specific about the types of impacts.	We appreciate the precision of review here. The text has been amended to make the basis of the comparison more exacting. Thank you.
3.5	Line 81+: The definition of a mine-town system is important and should be summarized in the text as early as possible. The closest to a clear definition is in the caption of Figure 1 but is still missing key information. The details are well described in lines 387+ in the Methods that the journal format puts at the end of the article, but leaves the reader unclear what they are when reading the main text.	See response to 1.2 above.
3.6	Additionally, the term “town” is a smaller population as defined in Supplemental Data 3 but unclear here that the “systems” part of the term “mine-town systems” includes the cities. Concept is fine but emphasizes the need for clearer definitions in the paper up front, since they are scattered across the text, captions, methods, and supplemental data.	Again, thank you. We have offered minor changes here to denote the distinction and relationship between towns and cities. We agree that clearer definitions are essential to readers extracting the fullest value from this research.
3.7	Line 120: The lack of identified mine-town systems in China, as well as Russia and India, in the top 20 seems like a serious data gap. Also, much of the concern of ETM mining has been for communities in Africa and South America and they do not seem to be on the list. This seems like enough of a gap that perhaps the paper should be recontextualized as a study of North America, Australia, and perhaps Europe. However, later in the paper, Figure 3 shows Russia, China, Africa, etc. So the geographic scope of this study needs to be clarified or made consistent.	The scope of the study is global, as emphasized in the title, abstract and across the manuscript. The data used for this study are without bias towards global north or south (see the method section). We recognise that the top mine-town systems in Tables 1-3 do not list the less developed countries. This has been rectified by the examples in Figures 5b, 6 and in the Supplementary Data 2.
3.8	Line 136: Some specialized coal mines may remain open for high value material such as metallurgical coal for steel production or coal reserves for backup power. So I think the authors can say nearly every open coal mine or functionally every coal mine will close but maybe not all. Results will be the same.	Thank you. We have incorporated this exact language.
3.9	Lines 210-220: Can this be balanced with any positive effects of mine closures, such as towns able to diversify and transform their economies? Many former coal mining towns in the western U.S. grew and benefited after their mines closed, although it can take decades.	We could have included a discussion about the positive effects of mine closures. However, this “balanced view” would need to be heavily qualified owing to the fact that most closures occur on an unplanned basis, and only after

		divestment options have been exhausted or offloading the asset as strategy failed. This leaves the area un-remediated and fully exposed to uncontrolled environmental impacts. I think the reviewer means to say that in isolated cases, areas formerly associated with mining can develop new industries and opportunities. It would be important to state that these opportunities emerge despite the closure of the mine, not because of it. With so many caveats our preference is to leave the text as is, using the data and findings we present.
3.10	Lines 237-239: Similar to previous comment, what are the negatives to communities of starting or expanding a mine, in addition to the eventual closure?	Mining projects at start up, expansion and at closure introduce enormous social, environmental and economic impacts. These impacts are well captured in the scholarly literature, including De Valck et al. 2021 ; Sovacool 2021 ; Crous et al. 2020 ; Lèbre et al. 2020 ; Lèbre et al. 2019 ; Haney and Shkaratan 2016 ; and Marais 2013 .
3.11	Lines 290-291: Will the bulk of mine-town systems affected by the energy transition really be in Canada or Australia, or is that an artefact of incomplete data sets? Vast amounts of mined materials that go into renewable energy come from China, as well as Africa and South America, so it is difficult to believe Canada will be most impacted. The data gaps are mentioned in lines 293-295 but they are significant enough that lines 290-291 seem like an overstatement that could be taken out of context.	We adjusted the paragraph by adding “As such, there may be more mine-town systems, particularly in Asia, Russia, Africa and South America than we present in our results. This points toward the need for local and national governments to collect more reliable data on resource inventories or make reporting more rigorous in Africa, South America and Asia, especially China.”
3.12	Line 345: Unclear what ‘recalcitrant’ means here, as this is the first and only time this term is used in this paper.	We have offered a brief follow up sentence explaining what is meant by ‘recalcitrant’.
3.13	Link 437-438. 200 km/2 hours seems excessive for commute time, especially when global medians are about 30 minutes. If 2 hours is viewed as a maximum it seems that the likelihood	See our response to comments 2.28 – 2.29.

	of a community being a mine-town would decline as distance increases past the median. Or perhaps this does not mean daily commutes? Authors need to define the basis for this assumption and how the distances were selected.	
3.14	Line 458+: Are artisanal mines included? These have some of the highest community impacts. While it may not be possible to include them, they should be mentioned in the limitations.	We have included the data limitation on artisanal and small-scale mining in the study limitations as the reviewer recommended.
3.15	Supplemental Data 2 maps. For large countries, such as Russia, China, and the U.S., the subnational data may be more important than the number of projects and settlements per country. A heat map of the cities would be more representative than country scale.	We have added Supplementary Data 5 where we present heatmaps of the mine-town systems potentially impacted by the coal phase-out and ETMs phase-in assumptions.

REVIEWER COMMENTS

Reviewer #1 (Remarks to the Author):

I sincerely appreciate the authors' work on the revised manuscript. I still have hesitations that the lack of socioeconomic considerations leaves the findings a bit flat. There is no recognition of disproportionate impacts by, say, race or gender. I find this problematic. However, the work is well done and the novel. So, I would recommend publication but would encourage the authors to explore this limitation further in the conclusion.

Reviewer #2 (Remarks to the Author):

Authors should be commended for their extensive revisions, which appear to have addressed the bulk of my comments (and those of the other reviewers, I believe). Although I am not a big fan of responses without signposting, I understand that significant progress has been made and I would be very happy to recommend publication, pending a few minor points.

In the abstract (previous point 2.3), the "extreme scenarios occurring concurrently" still reads off. I understand the point that thermal coal and energy transition metals projects are typically not co-located, and the vast majority of mine-town systems will not be affected by both transitions, but the reader of the abstract will be left confused.

In the Introduction, on the study's scope (previous point 2.4), I understand the aim is already there (last paragraph of introduction) but not spelled out explicitly. The addition on mapping dynamics is welcomed, but I would suggest that this sentence be rephrased to highlight that part, e.g. "First, our work establishes the basis for mapping likely phase-out and phase-in dynamics, by incorporating global resource inventories for coal and energy transition metals required for renewable technologies, thereby spanning both sides of the energy transition". In any case, it's hard to conceive the term "likelihood phase-out and phase-in dynamics" and I can only assume the authors mean "likely" here.

Moving onto Results and Discussion, on defending that current ETM projects will likely be insufficient (previous point 2.10), I understand the point made by the authors, but I do not see where in Methods this is addressed/tracked, nor why it has not been pinpointed in the main text as suggested, to facilitate readers.

At this point, I would strongly encourage authors to use lines and/or some sort of signposting in their responses, contrary to their first revision – which is common practice to facilitate reviewers.

Supplementary Data 3 (previous point 2.19): thank you for correcting as suggested. Please also correct for Zinc (directly impacted population figure).

On population evolution (previous point 2.21), shouldn't authors be making a case for possible understatement of their findings (assuming population keeps growing), as the population data used is today's? Unless I did not fully grasp their response.

Many thanks for adding the definition of a mine-town system (previous point 2.23). Just pointing out that I was referred to response 1.2, where there was no signposting and the definition is not tracked – following up on my signposting comment, authors are kindly requested to use tracked changes for all revisions (which was not the case at all in this version, as much of the plain and expectedly unchanged text in reality was revised/moved/etc.).

Excellent work on Supplementary Data 2. I would only suggest, in the Resource Inventories bit for all three (or, basically for Salta and Mackay), to clearly list Resource Inventory II projects below Resource Inventory III, as in "of which..." not to confuse the reader with total and broken-down figures.

Very informative Supplementary Data 4-5. Please fix Supplementary Data 4 caption. In

Supplementary Data 5 (heatmap), there appear to be coloured regions expanding to the sea (e.g., above Spain, below Hanoi, etc. for coal phase-out) in some cases; please ensure colours are correctly mapped, as this is not something the reader/reviewers can easily validate.

Again, the authors should be praised for their revisions and efforts to address all points raised. From my side, once the above minor revisions are carried out, I believe the manuscript can be accepted for publication.

A.N.

Reviewer #3 (Remarks to the Author):

My comments have been addressed. Thank you for the careful consideration. Recommended for publication with the usual editorial review.

Reviewer Response table for NCOMMS-22-19417

Reviewer 1

#	Reviewer Comment	Authors' response
1.1	I sincerely appreciate the authors' work on the revised manuscript. I still have hesitations that the lack of socioeconomic considerations leaves the findings a bit flat. There is no recognition of disproportionate impacts by, say, race or gender. I find this problematic. However, the work is well done and the novel. So, I would recommend publication but would encourage the authors to explore this limitation further in the conclusion.	Thank you for the appreciative and helpful comments that guided our revisions. We have added a further point on the socioeconomic considerations in the Conclusion (see l. 391-5).

Reviewer 2

#	Reviewer Comment	Authors' response
2.1	Authors should be commended for their extensive revisions, which appear to have addressed the bulk of my comments (and those of the other reviewers, I believe). Although I am not a big fan of responses without signposting, I understand that significant progress has been made and I would be very happy to recommend publication, pending a few minor points.	Thank you for your helpful comments. We did a comprehensive re-work of the document, and we thought that detailed tracking and signposting would not be helpful. Our apologies as it seems that this was not the case. In this round, we have tracked all changes by noting the line numbers where changes have been made.
2.2	In the abstract (previous point 2.3), the “extreme scenarios occurring concurrently” still reads off. I understand the point that thermal coal and energy transition metals projects are typically not co-located, and the vast majority of mine-town systems will not be affected by both transitions, but the reader of the abstract will be left confused.	Thank you we have used some of your helpful wording here. “While it is unlikely that extreme scenarios will occur concurrently, an empirical basis for examining social, economic, and environmental change as global markets adjust to the decline in thermal coal is established.” was rewritten to: “It is unlikely that extreme transition scenarios will occur concurrently. Thermal coal and energy transition metals projects are typically not co-located, and so the vast majority of mine-town

		systems will not be affected by both transitions at the same time. The mine-town systems approach establishes an empirical basis for examining the spatial extent of the transition, and offers policy makers a means for considering the demographic effects of the energy “switch”, and the associated infrastructure and economic planning needed as global markets adjust to the decline in thermal coal and the ramp up of other mining activities. (l. 15-21).
2.3	In the Introduction, on the study’s scope (previous point 2.4), I understand the aim is already there (last paragraph of introduction) but not spelled out explicitly. The addition on mapping dynamics is welcomed, but I would suggest that this sentence be rephrased to highlight that part, e.g. “First, our work establishes the basis for mapping likely phase-out and phase-in dynamics, by incorporating global resource inventories for coal and energy transition metals required for renewable technologies, thereby spanning both sides of the energy transition”. In any case, it’s hard to conceive the term “likelihood phase-out and phase-in dynamics” and I can only assume the authors mean “likely” here.	We have rewritten the sentence in l. 80-83 as suggested: “First, our work establishes a basis for mapping likely phase-out and phase-in dynamics of mineral resource extraction. Our work spans both sides of the energy transition by incorporating resource inventories for coal on the one hand, and the energy transition metals required for renewable technologies on the other.”
2.4	Moving onto Results and Discussion, on defending that current ETM projects will likely be insufficient (previous point 2.10), I understand the point made by the authors, but I do not see where in Methods this is addressed/tracked, nor why it has not been pinpointed in the main text as suggested, to facilitate readers.	We addressed this comment in section Impacts of shifting commodity markets, l. 167-170.
2.5	At this point, I would strongly encourage authors to use lines and/or some sort of signposting in their responses, contrary to their first revision – which is common practice to facilitate reviewers.	For each change made in this revision, we refer to related lines in the manuscript.
2.6	Supplementary Data 3 (previous point 2.19): thank you for correcting as suggested. Please also correct for Zinc (directly impacted population figure).	The figure for Zinc has been corrected.
2.7	On population evolution (previous point 2.21), shouldn’t authors be making a case for possible understatement of their findings (assuming population keeps growing), as the population data used is today’s? Unless I did not fully grasp their response.	We have reflected on this comment in Limitations, see added text in l. 562-565.

2.8	Many thanks for adding the definition of a mine-town system (previous point 2.23). Just pointing out that I was referred to response 1.2, where there was no signposting and the definition is not tracked – following up on my signposting comment, authors are kindly requested to use tracked changes for all revisions (which was not the case at all in this version, as much of the plain and expectedly unchanged text in reality was revised/moved/etc.).	Point taken.
2.9	Excellent work on Supplementary Data 2. I would only suggest, in the Resource Inventories bit for all three (or, basically for Salta and Mackay), to clearly list Resource Inventory II projects below Resource Inventory III, as in “of which...” not to confuse the reader with total and broken-down figures.	Thank you for the suggestion. The list of Resource Inventories has been edited per your suggestion.
2.10	Very informative Supplementary Data 4-5. Please fix Supplementary Data 4 caption. In Supplementary Data 5 (heatmap), there appear to be coloured regions expanding to the sea (e.g., above Spain, below Hanoi, etc. for coal phase-out) in some cases; please ensure colours are correctly mapped, as this is not something the reader/reviewers can easily validate.	The caption of Supplementary Data 4 has been edited and the heatmap in Supplementary Data 5 corrected as suggested.
2.11	Again, the authors should be praised for their revisions and efforts to address all points raised. From my side, once the above minor revisions are carried out, I believe the manuscript can be accepted for publication. A.N.	Thank you. Your comments are appreciated and reflected in the improved manuscript.

Reviewer 3

#	Reviewer Comment	Authors' response
3.1	My comments have been addressed. Thank you for the careful consideration. Recommended for publication with the usual editorial review.	We would like to thank the reviewer for the comments that helped us to improve the manuscript.